# Unfolding Generative Flows with Koopman Operators: Fast and Interpretable Sampling

## Abstract

Continuous Normalizing Flows (CNFs) enable elegant generative modeling but remain bottlenecked by slow sampling: producing a single sample requires solving a nonlinear ODE with hundreds of function evaluations. Recent approaches such as Rectified Flow and OT-CFM accelerate sampling by straightening trajectories, yet the learned dynamics remain nonlinear black boxes, limiting both efficiency and interpretability. We propose a fundamentally different perspective: globally linearizing flow dynamics via Koopman theory. By lifting Conditional Flow Matching (CFM) into a higher-dimensional Koopman space, we represent its evolution with a single linear operator. This yields two key benefits. First, sampling becomes one-step and parallelizable, computed in closed form via the matrix exponential. Second, the Koopman operator provides a spectral blueprint of generation, enabling novel interpretability through its eigenvalues and modes. We derive a practical, simulation-free training objective that enforces infinitesimal consistency with the teacher's dynamics and show that this alignment preserves fidelity along the full generative path, distinguishing our method from boundary-only distillation. Empirically, our approach achieves competitive sample quality with dramatic speedups, while uniquely enabling spectral analysis and editing-control of generative flows.

## 1 Introduction

While classic generative models like VAEs Kingma & Welling (2014) and GANs Goodfellow et al. (2014) offer fast, interpretable sampling, they have been surpassed in sample fidelity by dynamical system-based approaches like Diffusion Models Ho et al. (2020); Song et al. (2020) and Continuous Normalizing Flows (CNFs) Chen et al. (2018). This leap in quality, however, comes at the cost of slow, iterative sampling and limited interpretability.

For both model families, sampling is an iterative and slow process. Diffusion models learn to iteratively denoise data and therefore require multiple evaluations to generate samples, while sampling CNFs requires solving an ODE. In the case of CNFs, recent work has focused on accelerating sampling, with approaches such as Rectified Flow (Liu et al., 2023a) and Optimal Transport Conditional Flow Matching (Tong et al., 2024; Pooladian et al., 2023) that learn straighter generative paths. These methods successfully reduce the computational cost of generation while maintaining similar fidelity; however, they do not address the sampling process's lack of interpretability. This flaw limits our ability to understand *how* the model generates data, trust its outputs, and meaningfully control the generation process.

In this work, we address the challenges of slow sampling and limited interpretability in generative models grounded in dynamical systems. We build on Koopman operator theory, a classical framework for linearizing complex dynamical systems (Koopman, 1931; Mezić, 2005; Brunton et al., 2022). Originally developed in the 1930s, this theory has seen a resurgence in recent years thanks to machine learning methods that learn finite-dimensional approximations of the operator from data (Brunton et al., 2022; Bevanda et al., 2021). Neural network–based approaches such as Koopman autoencoders (Lusch et al., 2018; Otto & Rowley, 2019; Azencot et al., 2020) have successfully learned linear embeddings for complex systems in fields like fluid dynamics (Rowley et al., 2009) and molecular dynamics (Klus et al., 2018). We apply this approach to the dynamics of a pre-trained CNF, learning a latent space in which the dynamics evolve linearly under a corresponding learned linear operator (Lusch et al., 2018; Azencot et al., 2020). This transformation provides two key advantages:

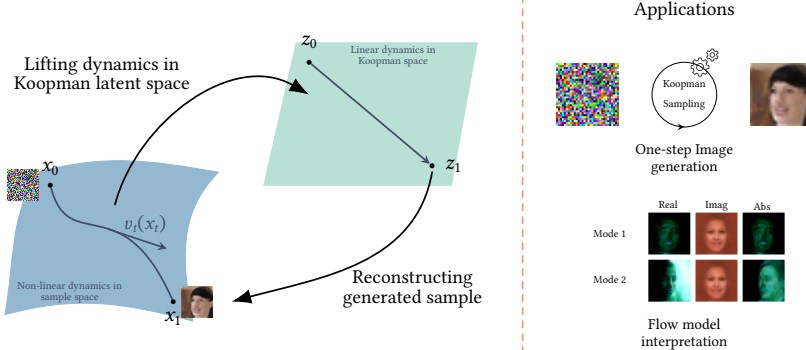

Figure 1: Overview of our approach: we propose to apply Koopman theory to the dynamics of generative modeling from continuous normalizing flow models. We learn a Koopman latent space and its linear dynamics from a given non-linear CNF model. This approach presents two direct applications: one-step sampling and flow model interpretability.

1. **Generative process decomposition:** The learned Koopman operator acts as an interpretable blueprint of the generative process. We show that either the learned canonical frame of the Koopman latent space, or the eigendecomposition of the Koopman operator reveal semantic components of the dynamics. This allows for an unprecedented analysis of how CNF models constructs data from noise.

2. **One-Step Analytical Sampling:** A direct consequence of this linearization is that the solution to the generative ODE becomes analytical, given by a matrix exponential. This allows us to map noise to a data sample in a single, parallelizable step, eliminating the iterative sampling cost entirely.

Our core contribution is a practical, simulation-free training objective that learns this Koopman representation. We theoretically prove that naïve supervision strategies yield suboptimal objectives and impractical training processes. Crucially, we derive an efficient supervision strategy that constrains the learned linear dynamics to stay consistent with the teacher model's vector field along the *entire* generative path. We show that this can be enforced while remaining simulation-free, inheriting the properties of the underlying Continuous Flow Matching model. This distinguishes our approach from standard distillation methods, that only match the start and end points of the trajectory, while incurring only a moderate additional computational cost. Specifically, our contributions are:

- We introduce a novel framework **for learning** a global Koopman linearization of the non-autonomous dynamics in Conditional Flow Matching models.

- We derive a practical, simulation-free training objective that enforces consistency along the full generative trajectory, yielding a **full** linearization rather than mere boundary-focused distillation.

- We demonstrate empirically that our method achieves competitive one-step sampling performance while uniquely enabling spectral analysis, disentangled generative control, and **improved robustness in** downstream tasks.

## 2 RELATED WORK

Our work connects four main areas: flow-based generative models, methods for accelerated sampling, Koopman operator theory for dynamical systems, and interpretability in generative modeling. We defer a formal introduction of Koopman operator theory to Section 3.2. For an overview of the field, we urge the interested reader to refer to the excellent introduction by Brunton et al. (2022).

## 2.1 FLOW-BASED GENERATIVE MODELS

Flow-based models learn an invertible mapping between a data distribution and a simple base distribution, offering tractable likelihoods (Dinh et al., 2014; 2017; Kingma & Dhariwal, 2018). Continuous Normalizing Flows (CNFs) parameterize this map as the solution to an ODE (Chen et al., 2018). Although powerful, training early CNFs was often unstable and computationally intensive. Conditional Flow Matching (CFM) represents a major step forward, providing a stable and efficient simulation-free training objective by regressing a neural network to a conditional vector field (Lipman et al., 2023; Tong et al., 2023; Liu et al., 2023b). However, while these models have achieved high accuracy for generative modeling, their sampling process remains inherently slow, opening the way for distilled models for faster sampling.

## 2.2 ACCELERATED AND ONE-STEP SAMPLING

The slow and iterative sampling of CNFs has motivated extensive research into acceleration. One popular direction, which includes Rectified Flow (Liu et al., 2023a) and OT-CFM (Pooladian et al., 2023), regularizes the learned ODE to have straighter trajectories, thus requiring fewer discretization steps. Another direction uses knowledge distillation to train a separate student model capable of single-step generation. This includes Consistency Models (Song et al., 2023) and other distillation techniques (Salimans & Ho, 2022; Luo et al., 2023; Liu et al., 2025). Although these methods achieve remarkable speed, they typically produce a compressed, black-box sampler that does not offer the interpretability or analytical control that our Koopman framework provides.

We also note that concurrently with our work, Berman et al. (2025) propose a Koopman-based generative model that learns a discrete-time Koopman operator, mapping noisy samples at $t = 0$ directly to target data at $t = 1$. While their approach is primarily positioned as an enhancement to diffusion models (though not exclusive to them), *our work focuses on conditional flow matching*, framing the problem as supervised learning of vector fields over time. In contrast to their discrete formulation, *we explicitly model the full continuous-time dynamics* by learning the Koopman generator, granting access to the entire latent flow from $t = 0$ to $t = 1$.

## 2.3 INTERPRETING AND EXPLAINING GENERATIVE MODELS

While methods exist for interpreting the latent spaces of classic models, such as VAEs and GANs, extending these powerful editing techniques to modern, iterative models like diffusion and flows has proven challenging due to their complex dynamics. Existing approaches for these models are often more complicated than the earlier methods Kwon et al. (2022); Yang et al. (2023); Meng et al. (2022); Kulikov et al. (2024), in addition to lacking the conceptual clarity of the latter. In contrast, our work offers a direct path to interpretability by learning a global linearization of the generative dynamics, which naturally yields a simple and editable latent space. A more detailed review of interpretability methods is provided in Appendix G.

# 3 MATHEMATICAL BACKGROUND

## 3.1 CONDITIONAL FLOW MATCHING

A Continuous Normalizing Flow (CNF) maps a prior distribution $p_0$ to a data distribution $p_1$ by solving the ODE

$$\frac{dx_t}{dt} = v_t(x_t), \text{s.t. } x_0 \sim p_0, x_1 \sim p_1 \tag{1}$$

, where $v_t$ is a time-dependent vector field Chen et al. (2018). A naive regression loss to learn $v_t$ is intractable, as both the true field $v_t$ and the marginal path distribution $p_t$ are unknown Lipman et al. (2023). Conditional Flow Matching (CFM) provides a tractable, simulation-free objective by regressing a neural network $v_\theta$ onto a *conditional* velocity field $u_t(x_t|x_1)$.

Sampling from a trained CFM model requires numerically integrating its ODE via $x_1 = x_0 + \int_0^1 v_\theta(s, x_s)ds$, a slow process with potentially many function evaluations Chen et al. (2018). However, if the dynamics were linear, i.e., of the form $\frac{dx_t}{dt} = Ax_t$, sampling would become a single,

analytical step: $x_t = e^{At}x_0$ that can be solved via matrix exponentiation. This vast efficiency gap motivates our core objective: to find a global linearization of the learned CFM dynamics.

## 3.2 KOOPMAN THEORY FOR AUTONOMOUS SYSTEMS

Koopman theory provides a powerful framework for globally linearizing nonlinear dynamical systems (Koopman, 1931; Mezić, 2005; Brunton et al., 2022). The central idea is to shift perspective from the finite-dimensional state space, where dynamics are nonlinear, to the infinite-dimensional space of functions - referred to as "observables" - where the dynamics become linear.

Formally, consider an autonomous dynamical system $\frac{dx_t}{dt} = v(x_t)$. This system induces a *flow map* $F_t$ that advances an initial state $x$ to its value at time $t$, namely $x_t = F_t(x)$, along the trajectories defined by $v$. Let $g : \mathbb{R}^d \to \mathbb{R}$ be an *observable function* on the state space. Given an initial state $x$, we define the *Koopman operator* $\mathcal{K}_t$ on the space of observables, denoted $\mathcal{G}(\mathbb{R}^d)$, which evolves observables along the trajectories generated by the vector field $v$:

$$\mathcal{K}_t g(x) := (g \circ F_t)(x) = g(F_t(x)) = g(x_t). \tag{2}$$

Koopman theory builds on the fact that this operator is trivially linear (regardless of the non-linearity of $F_t$) due to the linearity of the composition of functions: $K_t(g_1 + g_2)(x) = (g_1 + g_2) \circ F_t(x) = g_1 \circ F_t(x) + g_2 \circ F_t(x) = \mathcal{K}_t g_1(x) + \mathcal{K}_t g_2(x)$, for all observables $g_1, g_2$.

Taking the Lie derivative, we can then define the **Koopman generator**, $\mathcal{L}$, such that $\mathcal{L}g := \lim_{t\to 0} \frac{\mathcal{K}_t g - g}{t}$, and one can show that Brunton et al. (2022)

$$\mathcal{L}g = \frac{dg}{dt} = \nabla_x g(x) \cdot v(x), \tag{3}$$

which is also trivially linear in $g$, leading to a linear equation on the space of observables. The operator and generator are related by the matrix exponential, $\mathcal{K}_t = \exp(t\mathcal{L})$. Finding $\mathcal{L}$ is the objective of Koopman theory.

In summary, the potentially complex and non-linear ODE Equation (1) on the finite-dimensional state space $\mathbb{R}^d$ can be expressed as a linear equation in another space, $\mathcal{G}(\mathbb{R}^d)$, which consists of scalar-valued functions defined on the state space. The practical challenge in Koopman theory is to find *invertible mappings* $f : \mathbb{R}^d \to \mathcal{G}(\mathbb{R}^d)$ that allow solving the linear equation in the observable space and then recovering the solution in the original state space. However, computing such a mapping is often intractable in practice due to the *infinite dimensionality* of $\mathcal{G}(\mathbb{R}^d)$.

A particular case arises when there exists an $m$-dimensional linear subspace of $\mathcal{G}(\mathbb{R}^d)$, $F = \text{span}\{g_i\}_{i=1}^m$, invariant under the linear operator $\mathcal{L}$. The action of the generator on $F$ can then be represented by a single finite-dimensional matrix $L \in \mathbb{R}^{m \times m}$. The dynamics on this space of observables can then be written as:

$$\frac{d\mathbf{g}_t}{dt}(x) = L\mathbf{g}_t(x), \tag{4}$$

where $\mathbf{g}_t(x) = [g_1(x_t), \ldots, g_m(x_t)]^\mathsf{T} \in \mathbb{R}^m$ are the *Koopman coordinates*, i.e., the values of the observables $\{g_i\}_{i=1}^m$ evaluated at the state $x_t$, where $x_t$ is the evolution of the initial state $x$ to time $t$ along the trajectories generated by the dynamics.

**Fast analytical integration with Koopman operator**  Thus, the general goal when applying Koopman theory to dynamical systems is to (1) identify a sufficiently expressive set of observables $\{g_i\}_{i=1}^m$ and (2) determine the Koopman generator matrix $L$ on this space of observables. With this in hand, we can build an invertible Koopman representation $g : \mathbb{R}^d \to \mathbb{R}^m$ that maps a state $x$ to its Koopman coordinates $\mathbf{g}(x)$. This enables us, given an initial state $x_0 \in \mathbb{R}^d$, to solve the ODE associated with a *nonlinear dynamical system* in a space where it evolves linearly, using the matrix exponential $\mathbf{g}_1 = e^L g(x_0) \in \mathbb{R}^m$. We can then recover the solution of the ODE in the original state space by applying the inverse map $x_1 = g^{-1}(e^L g(x_0)) \in \mathbb{R}^d$.

**Mode decomposition of Koopman operator**  Another appeal of the Koopman theory is that it exposes an *interpretable* structure of the ODE, as we can decompose the different modes of the linear Koopman operator $L$. Intuitively, Koopman theory serves as a nonlinear analogue of Fourier

analysis: just as Fourier modes decompose signals into orthogonal oscillatory components, Koopman eigenfunctions decompose dynamics into independent modes with specific growth rates. We employ the *real Schur decomposition*:

$$L = QTQ^\top, \tag{5}$$

which represents each conjugate pair as a real $2 \times 2$ block and each real eigenvalue as a $1 \times 1$ block. A key property of the Koopman representation is that in Schur coordinates $y_t = Q^\top z_t$, the matrix exponential decomposes into independent modes. For a real eigenvalue $\lambda$, the corresponding $1 \times 1$ block yields an exponential mode $y(t) = e^{\lambda t} y(0)$, while $2 \times 2$ blocks of the form

$$\begin{pmatrix} \sigma & \omega \\ -\omega & \sigma \end{pmatrix} \tag{6}$$

yield planar spirals $y(t) = e^{\sigma t} R(\omega t) y(0)$ with radial rate $\sigma$ and rotation frequency $\omega$. Importantly, in both cases, the norm of each component grows according to a predictable exponential rate: $e^{\lambda t}$ or $e^{\sigma t}$. This provides a canonical *ordering* to all the modes (akin to ordering Fourier modes by frequency).

## 4 METHODOLOGY AND THEORETICAL RESULTS

Our objective is to learn a Koopman representation for a pre-trained CFM model, specified by its vector field $v_t$. This involves learning an encoder $g_\phi$ for the Koopman representation that linearizes the dynamics, a generator matrix $L$, and a decoder $g_\psi^{-1}$ that maps back to the state space. Here $\phi$ and $\psi$ are the learnable parameters of the corresponding neural networks. Several additional challenges arise compared to previous neural Koopman-based approaches Lusch et al. (2018):

1. CFM dynamics are non-autonomous (explicitly time-dependent), whereas classic Koopman theory applies to autonomous systems.

2. The training objective for the Koopman representation must be tractable, ideally inheriting the simulation-free nature of CFM.

3. The learned observables $g$ must be expressive enough to capture the dynamics and allow for accurately generated samples.

### 4.1 ADAPTING KOOPMAN THEORY TO NON-AUTONOMOUS DYNAMICS

**Time dependence trick.** As mentioned above, Koopman theory applies to autonomous dynamics, where the velocity $v(x_t)$ does not depend on the time. We can address this time-dependence of $v_t(x_t)$ by using a standard trick in system dynamics literature (Strogatz (2000), Chap 1.): we augment the state space to include time. The state becomes $y_t = (t, x_t)$, and the dynamics are defined on this augmented space with respect to a new external time parameter $\tau$:

$$\frac{dy}{d\tau} = \frac{d(t, x_t)}{d\tau} = [1, \ v_t(x_t)]. \tag{7}$$

Our observables are now functions of both space and time, $g(t, x)$. A crucial detail, however, is how we parameterize the linear dynamics on this augmented state to ensure the time variable evolves correctly (i.e., $\dot{t} = 1$).

**Affine lift for time evolution.** To enforce the constraint $\dot{t} \equiv 1$, we use an *affine lift*. The state is augmented with a constant bias coordinate to become $\mathbf{z}_t = [1, \ t, \ g(t, \ x)]^T$. For the dynamics $\dot{\mathbf{z}} = L\mathbf{z}$ to satisfy the physical constraints $\dot{1} = 0$ and $\dot{t} = 1$ for all states, the generator $L$ is uniquely constrained to adopt a block structure. The precise parameterization of $L$ is available in the appendix.

### 4.2 LEARNING KOOPMAN DYNAMICS

Given a pre-trained CFM teacher network $v_t$, our main goal is to learn observable functions $\{g_i\}_{i=1}^m$ that span a finite-dimensional subspace *invariant under* the Koopman generator $\mathcal{L}$ *associated with* the dynamics $v_t$, and to learn the corresponding generator on this space. We learn the observables with an encoder $g_\phi$ that maps an initial state $x \in \mathbb{R}^d$ to its Koopman coordinates at time $t$, $\mathbf{g}_t(t, \ x) = [g_1(t, \ x_t), \ \ldots, \ g_m(t, \ x_t)]^T \in \mathbb{R}^m$. We also learn the Koopman generator on this space as a dense

matrix $L \in \mathbb{R}^{m \times m}$. To recover the solution of the ODE in the original state space and ensure the learned linear dynamics correspond to the *underlying nonlinear dynamics*, we also learn a decoder network $g_\psi^{-1}$ that maps the Koopman coordinates $\mathbf{g}_t(x)$ back to the state $x_t$ at time $t$.

We generate noise and target-data pairs $(x_0, x_1)$ using the pretrained CFM model, and aim to learn the following mapping:

$$x_t \simeq g^{-1}(e^{tL}g(0, x_0)).$$

**Training loss** Our training objective is as follows:

$$\mathcal{L}_{\text{train}} = \lambda_{\text{phase}}\mathcal{L}_{\text{phase}} + \lambda_{\text{target}}\mathcal{L}_{\text{target}} + \lambda_{\text{recon}}\mathcal{L}_{\text{recon}} + \lambda_{\text{cons}}\mathcal{L}_{\text{cons}}.$$

The first two terms ensure that the integrated linear dynamics map the start of a trajectory to its end in the Koopman space (phase loss):

$$\mathcal{L}_{\text{phase}} = \mathbb{E}_{(x_0, x_1)} \left\| e^L g_\phi(0, x_0) - g_\phi(1, x_1) \right\|^2, \tag{8}$$

and in the state space (after decoding - target loss):

$$\mathcal{L}_{\text{target}} = \mathbb{E}_{(x_0, x_1)} \left\| g_\psi^{-1} \left( e^L g_\phi(0, x_0) \right) - x_1 \right\|^2, \tag{9}$$

The third term encourages that we can retrieve the final state with the decoder:

$$\mathcal{L}_{\text{recon}} = \mathbb{E}_{x_1} \left[ d_{\text{Image}} \left( g_\psi^{-1} \left( g_\phi(1, x_1) \right), x_1 \right) \right] \tag{10}$$

where $d_{\text{Image}}$ is a distance measure on the image space, such as MSE or LPIPS Zhang et al. (2018). The reconstruction loss is particularly important due to an inherent non-identifiability in the Koopman representation, as formalized in the proposition below. This term allows us to find, among the space of Koopman linearizing coordinate systems, the decodable ones.

We choose to only decode at $t = 1$ for those reasons: first, learning to reconstruct random noise may affect the capacity of the decoder to reconstruct images faithfully. Second, by not reconstructing intermediary states from observables, we give more flexibility to the encoder and generator to learn the proper Koopman representation space that manages to linearize the dynamics.

**Proposition 1** (Non-identifiability up to linear transformation). *The Koopman observable coordinates $g$ are identifiable only up to an arbitrary invertible linear transformation $M$. If the pair $(g, L)$ satisfies the consistency and phase objectives, so does the transformed pair $(M^{-1}g, M^{-1}LM)$.*

**Corollary 1.1.** *A reconstruction loss of the form $\|g^{-1}(g(t, x)) - x\|^2$, with a fixed decoder $g^{-1}$, breaks this invariance. It "fixes the gauge" by selecting the specific coordinate system that the chosen decoder can successfully map back to the data space.*

The proof is provided in Appendix A. This result motivates the necessity of $\mathcal{L}_{\text{recon}}$ to obtain a unique and useful representation.

Finally, the consistency loss forces the dynamics in the learned latent space to be governed by the linear generator $L$, by adapting Equation (3) to our problem:

$$\mathcal{L}_{\text{cons}} = \mathbb{E}_{t, x_t \sim p_t(x_t)} \left\| L g_\phi(t, x_t) - \nabla_x g_\phi(x_t) v_t(x_t) \right\|^2 \tag{11}$$

### 4.3 EFFICIENT DYNAMICS LEARNING

One might notice that, similarly to the CNF loss, the consistency loss $\mathcal{L}_{\text{cons}}$ is intractable, as it would require sampling from the path distribution $x_t \sim p_t(x_t)$. A first solution would be to generate full trajectories $(x_t)_t$, but this would pose both discretization and scale problems for storing the pre-computed trajectories. Another solution is to hope to substitute the marginal velocity $v_t(x_t)$ with the conditional velocity $u_t(x_t|x_1)$ and sample from the tractable $p_t(x_t|x_1)$, mirroring the CFM training strategy. However, as the following proposition shows, these two objectives are not equivalent when learning the encoder $g$.

**Proposition 2** (Marginal vs. Conditional Objectives). *Let $\mathcal{L}_{marg}$ be the desired consistency loss evaluated over the marginal distribution $p_t(x_t)$, and let $\mathcal{L}_{cond}$ be the tractable alternative evaluated using conditional samples and velocities. The two objectives are related by:*

$$\mathcal{L}_{cond} = \mathcal{L}_{marg} + \Delta(g) \tag{12}$$

*where $\Delta(g) = \mathbb{E}_{t, x_1, x_t} \left\| \nabla_x g(t, x_t)(u_t(x_t|x_1) - v_t(x_t)) \right\|^2 \geq 0$.*

The proof is provided in Appendix B. Because of the positive, $g$-dependent term $\Delta(g)$, minimizing $\mathcal{L}_{\text{cond}}$ will not necessarily minimize $\mathcal{L}_{\text{marg}}$.

Fortunately, as we have a pre-trained CFM model, the marginal velocity field $v_t(x_t)$ is known. This allows us to formulate a practical estimator for the true marginal loss, as stated in the following proposition.

**Proposition 3** (Practical Estimator for the Consistency Loss). *Given that the marginal path distribution $p_t(x_t)$ is defined as $p_t(x_t) = \int p_t(x_t|x_1)q(x_1)dx_1$, the marginal consistency loss $\mathcal{L}_{cons}$ can be estimated tractably using samples from the data distribution $q(x_1)$ and the conditional path $p_t(\cdot|x_1)$ as follows:*

$$\mathcal{L}_{cons} = \mathbb{E}_{t,\ x_1 \sim q_1,\ x_t \sim p_t(\cdot|x_1)} \left\| Lg_\phi(t,\ x_t) - \nabla_{\boldsymbol{x}} g_\phi(x_t)v_t(x_t) \right\|^2 \tag{13}$$

The proof is provided in Appendix C. This result is key: it allows us to **optimize the correct marginal objective using the same efficient, simulation-free sampling strategy** as CFM training, bypassing the need to compute and store full ODE trajectories.

Moreover, this loss is a key distinction of our method. Most single-step distillation-based generative models Song et al. (2023) focus on learning a direct mapping $\mathcal{D} : x_0 \mapsto x_1$ that minimizes a boundary-condition loss, like $\|\mathcal{D}(x_0) - x_1\|^2$. However, by focusing on endpoints, the distillation completely ignores the dynamics of the generative ODE. An infinite number of vector fields can satisfy the boundary conditions. In contrast, our approach seeks to perform a true *linearization* of the **full dynamics**. The inclusion of the infinitesimal consistency loss, $\mathcal{L}_{\text{cons}}$, forces our Koopman representation to remain faithful to the teacher's dynamics **at every point along the trajectory**.

## 4.4 GLOBAL LINEARIZATION AS AN INTERPRETABILITY AND CONTROL TOOL

As mentioned above, in addition to distilling a teacher model for faster sampling, we also aim to expose an *interpretable* structure within generative dynamics by leveraging Koopman operator theory. We highlight the value of our approach, as a tool to shed light on the underlying dynamics, as well as to *direct* the behavior of the teacher model.

**Image and mode inversion** A first step in interpreting Koopman modes is to un-lift them to the CFM dynamics. First, we highlight that our analytical sampling allows us to invert any image $x$ into the noise space, a task that is generally non-trivial for nonlinear generators and often requires specialized methods (e.g., Mokady et al. (2023) for diffusion models). We do so by computing the corresponding latent noise $g(0, x_0) = \exp(-L)g(1, x)$ and optimizing noise in pixel space which reproduces the latent. We detail and demonstrate some inversion examples in the supplementary material F.1.

We can then un-lift any mode $v_k$ into the pixel space. We do this by solving an inverse problem: Let $x_0 \sim p_0$ be a sample noise, $\mathbf{v}_i$ a Koopman mode. We search $x_{\text{pert}}^i$, such that:

$$x_{\text{pert}}^i = \arg\min_x ||g_\phi(0, x_0 + x) - g_\phi(0, x_0) + \alpha \mathbf{v}_i||^2. \tag{14}$$

**Class-conditioned spectral signatures** Let a dataset $D = \{x_i\}$ and $D_c = \{x_i(c)\}$. We encode each image and project it onto Koopman modes, giving coefficients $\alpha_i(k) = |\langle \phi_k, z_i \rangle|$ and $\alpha_i(k, c) = |\langle \phi_k, z_i(c) \rangle|$. We then compute the dataset and class-averaged responses

$$\bar{\alpha}(k) = \frac{1}{|D|} \sum_{i \in D} \alpha_i(k), \bar{\alpha}(k, c) = \frac{1}{|D_c|} \sum_{i \in D_c} \alpha_i(k, c), \tag{15}$$

and define the per-class transfer function

$$H(|\lambda|, c) = \bar{\alpha}(|\lambda|, c)/\bar{\alpha}(|\lambda|). \tag{16}$$

This measures how each class amplifies or suppresses modes of a spectral magnitude. By looking at which modes correspond to the highest class spectral deviation, we can understand which modes are common to images and which ones handle class-specific features.

**Semantic mode discovery** Given an image $x$, we perturb its lifted representation $z$ as $z'_k = z + \alpha v_k$. We measure the CLIP Radford et al. (2021) - a common embedding space for text and images - similarity between the decoded image and some selected attribute $\beta$ prompts $p_\beta$. We define the

*coherence* $C_k^\beta$ between a mode $v_k$ and an attribute $\beta$ as the sign consistency of similarity changes across test images:

$$C_k^\beta = \frac{1}{N} \sum_{i=1}^{N} \text{sign}(\langle \text{CLIP}(z_k'), \text{CLIP}(p_\beta) \rangle - \langle \text{CLIP}(z), \text{CLIP}(p_\beta) \rangle) \tag{17}$$

We can then select modes $v_i$ with the highest coherence for different attributes, allowing semantic editing of the images, both in the Koopman space and in the image space with the un-lifted modes $x_{\text{pert}}^i$.

**Insights on Teacher Training** We use our Koopman framework to probe how the CFM teacher acquires its dynamics during training. We compare the modes $v_i^l, v_j^{\text{full}}$ at different training stages $l$ with the modes of the the fully trained teacher by computing their similarity $S_{ij}^{l,\text{full}}$, and further their cumulative similarity $c_s(k)$ to the full matrix $v_{\text{full}}$:

$$S_{ij}^{l,\text{full}} = \left| \langle v_i^{l,\dagger} v_j^{\text{full}} \rangle \right| / \left( \|v_i^l\| \|v_j^{\text{full}}\| \right), \quad c_s(k) = \frac{1}{k} \sum_{i=1}^{k} S_{ii}^{l,\text{full}} \tag{18}$$

The full similarity matrix and the cumulative similarity at different stages indicate how much the teacher has learned compared to the final teacher, obtained insights are detailed in E.2.

# 5 EXPERIMENTS

To validate our framework, we investigate three key questions: (1) Can our one-step sampler achieve competitive generative quality? (2) Is the infinitesimal consistency loss ($\mathcal{L}_{cons}$) crucial for learning an interpretable linearization, as opposed to a simple boundary-matching distillation? (3) Does this learned structure lead to a more robust and functionally useful model? Our experiments show that while a simple distillation model can achieve a competitive FID Heusel et al. (2017) score, only the model trained with $\mathcal{L}_{cons}$ learns a disentangled, editable, and robust generative process.

## 5.1 EXPERIMENTAL SETUP

**Datasets and Teacher Model.** We evaluate on MNIST LeCun et al. (2010), CIFAR-10 Krizhevsky et al. (2009), and a 32x32 downsampled version of the FFHQ face dataset Karras et al. (2019). Our teacher is a pre-trained Optimal Transport Conditional Flow Matching (OT-CFM) model with a U-Net architecture. For boundary-based losses ($\mathcal{L}_{\text{target}}, \mathcal{L}_{\text{phase}}, \mathcal{L}_{\text{recon}}$), we use 1 million pre-generated $(x_0, x_1)$ pairs from the teacher network.

**Koopman-CFM Architecture.** Our model consists of an encoder ($g_\phi$) and decoder ($g_\psi^{-1}$), both using a `SongUNet` architecture Karras et al. (2022), which map to and from a 1024-dimensional latent space. The dynamics are governed by a learned affine linear generator ($\tilde{L}$).

**Training and Baselines.** We train for 800,000 iterations using the Adam optimizer Kingma & Ba (2017). Our primary baseline is an ablation of our own model trained without the consistency loss ($\mathcal{L}_{cons} = 0$), which reduces it to a standard distillation model.

## 5.2 GENERATION QUALITY

We evaluate sample quality using the Fréchet Inception Distance (FID), shown in Table 1. Our full Koopman-CFM model with consistency achieves competitive performance. Interestingly, the model trained without consistency achieves a slightly superior FID on FFHQ (7.5 vs. 8.5 for the teacher). This suggests that, when only constraining the endpoints, the distillation model is free to find a combination of paths and latent space that is easier to learn. As mentioned above, however, such a model is not guaranteed to replicate the trajectories of the teacher model. We provide uncurated generated examples with the consistency trained model in the appendix Section D.

## 5.3 ABLATION

**Koopman space dimension.** As shown in Figure 2, the Koopman dimension of 1026 (1024+2) is optimal for the generation quality. Notably, increasing the dimension to 1026 does not affect the quality with potential instabilities of the Koopman sampling components, such as the exponentiation.

Table 1: FID (↓) and sampling time (s/img, ↓) on three benchmark datasets. Our Koopman formulation achieves competitive or superior generation quality while enabling fast inference. Baselines are trained under identical preprocessing for fair comparisons. ♯ Indicates reproduction. Rectified Flow uses 2RF training and 1-step distillation.

| Method | NFE | MNIST | FFHQ | CIFAR-10 | Sampling Time (ms/img) |
|---|---|---|---|---|---|
| Koopman (ours, w/ consistency) | 1 | 7.1 | 10.1 | 17.4 | 37.2 |
| Koopman (ours, w/o consistency) | 1 | 6.4 | 7.5 | 14.1 | 37.2 |
| OT-CFM | 1 | 181 | 149 | 226 | 7.1 |
| OT-CFM | 3 | 28.1 | 51 | 59.3 | 25.2 |
| OT-CFM | 5 | 12.5 | 31.4 | 31.5 | 41.1 |
| OT-CFM | 25 | 4.4 | 11.6 | 12.3 | 209 |
| OT-CFM (Tong et al. (2024)) | 100 | 1.9 | 8.5 | 7 | 849 |
| Rectified Flow (Liu et al. (2023a)) | 1 | ♯ 1.76 | ♯ 4.23 | 4.85 | ♯ 24.9 |
| MeanFlow (Geng et al. (2025)) | 1 | ♯ 4.03 | ♯ 3.34 | ♯ 3.59 | ♯ 22.5 |

**Impact of consistency on trajectories.** We measure how $\mathcal{L}_{cons}$ affects the capacity of the model to reproduce the teacher's dynamics. To test this, we encode a teacher's trajectory $\{x_t\}_{t\in[0,1]}$ in the latent space and compare this ground truth path $z_t = g_\phi(t, x_t)$ against the analytical linear trajectory from our model, $\tilde{z}_t = \exp(\tilde{L}t)\tilde{z}_0$. We show the results in Table 2, with more details in the Appendix C.2. The trajectories are significantly better when using the consistency loss.

**Dimension scalability** Results in Table 1 show that we can learn Koopman representations with increasing variability, from the relatively simple MNIST to more complex datasets like FFHQ and CIFAR-10. To assess the dimension scalability, we trained a Koopman generator on FFHQ images of dimension 64x64. We obtain a FID of **13.4**, showing similar results when the dimension increases.

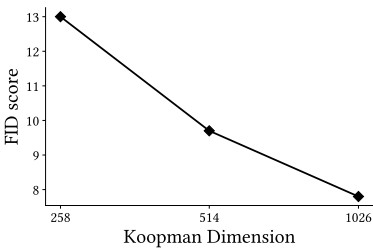

Figure 2: FID score as a function of Koopman dimension on the FFHQ dataset. The higher the dimension, the lower the FID.

| Dataset | Mean MSE |
|---|---|
| FFHQ (w/ consistency) | $\mathbf{5.0 \times 10^{-6}}$ |
| FFHQ (w/o consistency) | $1.30 \times 10^{-3}$ |
| CIFAR-10 (w/ consistency) | $\mathbf{1.0 \times 10^{-5}}$ |
| CIFAR-10 (w/o consistency) | $1.74 \times 10^{-3}$ |

Table 2: Mean, standard deviation of MSE between CFM trajectories and predicted Koopman trajectories. The consistency-trained model consistently outperforms the distilled model for trajectory fidelity

**Quality of the Koopman latent space.** We also provide a qualitative evaluation of the Koopman latent space. Namely, we borrow from the GAN literature and search for semantic directions in the latent space, such as glasses or gender. To find these directions in the latent space, we classify the dataset with attributes' CLIP (Radford et al. (2021)) embedding similarity and compute the mean and difference with relevant latents, see Appendix F.2. for more details. Given a semantically coherent mode, we invert it to the image space and compare the quality of the semantic editing.

As shown in Figure 3, furthermore in Appendix F.3., both latent spaces provide semantic directions, as expected from the faithful reconstruction of images. However, only the consistency-trained model transfers cleanly to the teacher, whereas the model without consistency introduces artifacts.

## 5.4 INTERPRETABILITY ANALYSIS

**Do Koopman Modes Encode Semantic Content?**

We compute the coherence of modes on the FFHQ dataset with four attributes, namely, *glasses, smile, brown* and *young*. Figure 4 compares the maximum coherence (Eq. 17) of models with and without consistency as well as the maximum mean CLIP difference, it also shows qualitative effect of the identified mode perturbation. The consistent model achieves near-perfect coherence for attributes like sunglasses (0.97) and brown hair (0.94), with variation magnitudes up to 24× larger. This

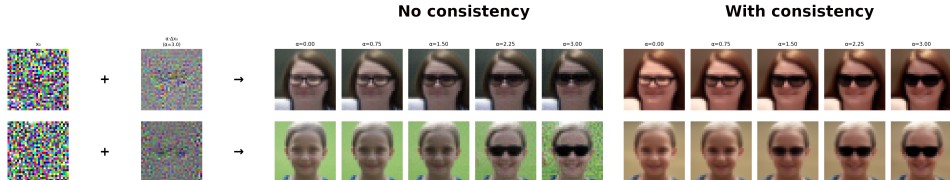

Figure 3: CFM-based semantic editing comparison. We port identified semantic directions from the Koopman latent space to the CFM noise-space via inversion as explained in F.3, here for sunglasses. Notably, we can see that recovered direction in the purely distilled model provides unreliable edits, as it comes with noticeable noisy artifacts, as opposed to our consistent model.

| Attr./Coherence | W/ | | W/o | |
|---|---|---|---|---|
| | Max. | Var. | Max. | Var. |
| Glasses | **.97** | **.046** | .48 | .002 |
| Smile | .52 | **.010** | .50 | .003 |
| Brown | **.94** | **.027** | .50 | .003 |
| Young | **.45** | **.007** | .32 | .001 |
| Avg. | **.72** | **.022** | .45 | .002 |

Figure 4: Semantic content of Koopman modes. **Left:** Maximal (Max.) and Variance (Var.) coherence of single-mode perturbations. **Center/Right:** Sunglasses mode perturbation with increasing effect (left→right within each panel). With consistency, the maximal "sunglasses" coherent eigenmode adds sunglasses while preserving identity. Without, perturbations produce negligible semantic change.

demonstrates that consistency is essential for learning modes that align with interpretable semantics. We provide more details in the Appendix E.4.

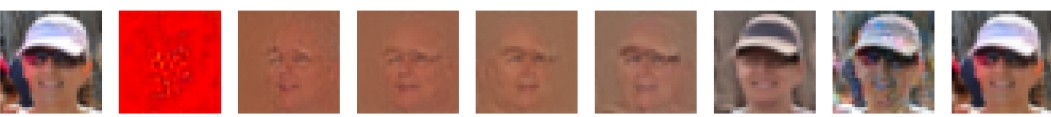

Figure 5: Progressive reconstruction of image as we progressively feed modes with ascending real eigenvalue part. The modes appear to act in a coarse-to-fine manner

**Class-conditioned spectral signatures**

We show in Fig. 6 the different transfer functions (Eq. 16) for each class of CIFAR-10. Notably, low-energy modes are largely shared across classes, while higher-energy modes differentiate them. This is further supported by observing Figure 5, where the important elements of the face appear before differentiating ones, such as the smile, hat, and background. This may shed light on our observations in Appendix E.2 on teacher training.

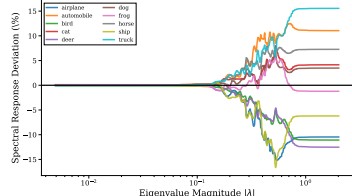

Figure 6: Per-class spectral deviation on CIFAR-10.

## 6 CONCLUSION AND DISCUSSION

We introduced a principled Koopman operator framework to linearize Conditional Flow Matching, achieving fast, one-step, and interpretable generative modeling on realistic image domains. Key challenges remain in scaling to high-resolution images, where the generator matrix becomes prohibitively large and its exponential can be numerically unstable. Future work should explore structured operator approximations and specialized matrix exponential algorithms to address these computational hurdles. Furthermore, we observe that the quality gap between our method and traditional CFM widens on more complex datasets, motivating a deeper theoretical investigation into the conditions under which CFM dynamics admit a finite-dimensional Koopman representation Iacob et al. (2023). Finally, the modality-agnostic nature of our framework opens exciting avenues for adapting this linearization approach to other data types, such as audio and 3D shapes.

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

APPENDIX

The supplementary materials below provide an expanded theoretical motivation, experimental details, and additional results that support and extend the main paper. Each section corresponds to specific elements of our method and results, with backward references to the main paper for clarity.

APPENDIX OVERVIEW

- **Section A: Theoretical Results and Proofs**
  In Section A we provide additional theoretical results and proofs, including the non-identifiability of Koopman coordinates, the non-equivalence of the conditional and marginal velocity field estimators and a tractable estimator for the marginal consistency loss.
- **Section B: Detailed Experimental Setup**
  In Section B we give details on the experimental setup, covering dataset preparation, architectures, hyperparameters, and computational resources.
- **Section C: Ablations**
  In Section C we present ablations, shedding light on the impact of loss terms on FID, the effect of Koopman dimension on FID, the role of consistency in trajectory fidelity, and the interpretability of modes with and without consistency.
- **Section D: Uncurated Samples and Sampling Speeds**
  In Section E we provide uncurated samples and wall-clock timings to further illustrate the speed–fidelity–interpretability tradeoff of our Koopman sampler.
- **Section E: Interpretability and Spectral Analysis**
  In Section E we analyze the learned Koopman representation, including mode structure with and without consistency, progressive coarse-to-fine reconstruction via eigenvalue ordering, class-conditioned spectral signatures, semantic mode discovery via CLIP coherence, and insights on teacher training dynamics.
- **Section F: Applications**
  In Section F we demonstrate practical applications of our framework, including image and mode inversion, discovering semantic directions in the Koopman latent space, noise engineering for CFM-based editing, and functional robustness on downstream tasks such as inpainting, super-resolution, and denoising.
- **Section G: Extended survey on interpretability of generative models**
  In Section G we provide a more extensive discussion on intepretrability of generative models.

Together, these sections provide a deeper understanding of our Koopman-CFM framework and support its efficiency, stability, and interpretability as claimed in the main paper.

## A   THEORETICAL RESULTS AND PROOFS

In this section we expand on the theoretical foundations introduced in Section 4 of the main paper. We provide detailed proofs of Theorem 1 and Propositions 1–3, which establish the non-identifiability of Koopman coordinates up to linear transformations and justify the inclusion of the reconstruction loss, as well as the derivation of a tractable marginal consistency objective. These results complement the main text by giving formal guarantees for the claims underlying our Koopman-CFM framework.

### A.1   PRELIMINARIES ON CFM

We remind here the main components of Conditional Flow Matching Tong et al. (2023), before deriving the proofs of our propositions. A Continuous Normalizing Flow (CNF) models the transformation from a prior distribution $p_0$ to a data distribution $p_1 = q_1$ via a probability path $p_t$. This path is induced by a time-dependent vector field $v_t$ through the ODE:

$$\frac{dx_t}{dt} = v_t(x_t), x_0 \sim p_0, x_1 \sim p_1 \tag{19}$$

where $x_t \in \mathbb{R}^d$ is a sample at time $t$. A naive objective to learn $v_t$ would be a regression loss:

$$\mathcal{L}_{\text{naive}} = \mathbb{E}_{t \sim U(0,1), x_t \sim p_t} \|v_\theta(t, x_t) - v_t(x_t)\|^2 \tag{20}$$

This objective is intractable because both the true vector field $v_t$ and the marginal path distribution $p_t$ are unknown. Conditional Flow Matching (CFM) circumvents this by defining a tractable conditional probability path $p_t(x_t|x_1)$ and its corresponding conditional vector field $u_t(x_t|x_1)$. The marginal velocity field $v_t$ can be expressed as an expectation over these conditional fields:

$$v_t(x_t) = \mathbb{E}_{x_1 \sim q(x_1|x_t)}[u_t(x_t|x_1)] = \int \frac{p_t(x_t|x_1)q(x_1)}{p_t(x_t)} u_t(x_t|x_1)dx_1 \tag{21}$$

Remarkably, CFM shows that minimizing a simulation-free objective based on the conditional velocity field is equivalent to minimizing the intractable marginal objective. The CFM loss is:

$$\mathcal{L}_{\text{CFM}} = \mathbb{E}_{t \sim U(0,1), x_1 \sim q_1, x_t \sim p_t(\cdot|x_1)} \|v_\theta(t, x_t) - u_t(x_t|x_1)\|^2 \tag{22}$$

While this makes training efficient, sampling requires solving the integral:

$$x_1 = x_0 + \int_0^1 v_\theta(s, x_s)ds \tag{23}$$

### A.2 PROOF OF THEOREM 1

*Proof.* Let the augmented state observable be $E(t, x) = [t, g(t, x)]^T$. We show that the objectives are invariant under the transformation $E \mapsto E_T = T^{-1}E$ and $L \mapsto L_T = T^{-1}LT$ for any invertible block-diagonal matrix $T = \text{diag}(1, M)$.

We use two facts. First, the chain rule implies that the Jacobian transforms as:

$$\mathrm{D}(E_T)[1, v_t] = \mathrm{D}(T^{-1}E)[1, v_t] = T^{-1}\mathrm{D}E[1, v_t]. \tag{J}$$

Second, the matrix exponential (and thus the flow) is conjugate under $T$:

$$\exp(\Delta t L_T) = T^{-1}\exp(\Delta t L)T. \tag{C}$$

**Infinitesimal Consistency.** The residual is $R_{\text{cons}} = \mathrm{D}E[1, v_t] - LE$. The transformed residual is:

$$R_{\text{cons},T} = \mathrm{D}E_T[1, v_t] - L_T E_T \overset{\text{(J),(C)}}{=} T^{-1}\mathrm{D}E[1, v_t] - T^{-1}LE = T^{-1}R_{\text{cons}}.$$

Thus, $R_{\text{cons}} = 0$ if and only if $R_{\text{cons},T} = 0$.

**Phase Loss.** The residual is $R_{\text{phase}} = E(1, x_1) - e^L E(0, x_0)$. The transformed residual is:

$$
\begin{aligned}
R_{\text{phase},T} &= E_T(1, x_1) - e^{L_T} E_T(0, x_0) \\
&= T^{-1}E(1, x_1) - (T^{-1}e^L T)(T^{-1}E(0, x_0)) \\
&= T^{-1}(E(1, x_1) - e^L E(0, x_0)) = T^{-1}R_{\text{phase}}.
\end{aligned}
$$

Again, the zero set of the loss is invariant. Since the norms of the residuals are scaled by the constant transformation $T^{-1}$, the set of global minimizers is preserved under this transformation. Therefore, the objectives only identify $g$ up to an invertible linear transformation $M$. $\square$

### A.3 PROOF OF PROPOSITION 1

*Proof.* To simplify the notation, let us define:

$$
\begin{aligned}
A(x_t) &= Lg(x_t) \\
B(x_t) &= \nabla g(x_t)\, v_t(x_t) \\
C(x_t, x_1) &= \nabla g(x_t)\, u_t(x_t \mid x_1)
\end{aligned}
$$

With this notation, the losses are $\mathcal{L}_{\text{marg}} = \mathbb{E}_{x_t \sim p_t}[\|A(x_t) - B(x_t)\|^2]$ and $\mathcal{L}_{\text{cond}} = \mathbb{E}_{x_1 \sim q, x_t \sim p_t(\cdot|x_1)}[\|A(x_t) - C(x_t, x_1)\|^2]$.

We expand the squared norms inside the expectations:

$$\mathcal{L}_{\text{marg}} = \int p_t(x_t) \left( \|A\|^2 - 2\langle A, B \rangle + \|B\|^2 \right) dx_t$$

$$\mathcal{L}_{\text{cond}} = \iint q(x_1) \, p_t(x_t \mid x_1) \left( \|A\|^2 - 2\langle A, C \rangle + \|C\|^2 \right) dx_t \, dx_1$$

We will now compare the terms of these two expansions one by one.

**(i) First Term ($\|A\|^2$):** The first term of $\mathcal{L}_{\text{cond}}$ is $\iint q(x_1) \, p_t(x_t \mid x_1) \, \|A(x_t)\|^2 \, dx_t \, dx_1$. Since $A(x_t)$ does not depend on $x_1$, we can use the law of iterated expectation or simply rearrange the integral:

$$\iint q(x_1) \, p_t(x_t \mid x_1) \, \|A(x_t)\|^2 \, dx_t \, dx_1 = \int \left( \int q(x_1) \, p_t(x_t \mid x_1) \, dx_1 \right) \|A(x_t)\|^2 \, dx_t$$

$$= \int p_t(x_t) \, \|A(x_t)\|^2 \, dx_t$$

This is identical to the first term of $\mathcal{L}_{\text{marg}}$.

**(ii) Cross Term ($-2\langle A, \cdot \rangle$):** The cross term of $\mathcal{L}_{\text{cond}}$ is $\iint q(x_1) \, p_t(x_t \mid x_1) \left( -2\langle A(x_t), C(x_t, x_1) \rangle \right) dx_t \, dx_1$. We analyze the integral:

$$\iint q(x_1) \, p_t(x_t \mid x_1) \, \langle A(x_t), C(x_t, x_1) \rangle \, dx_t \, dx_1$$

$$= \int \left\langle A(x_t), \int q(x_1) \, p_t(x_t \mid x_1) \, C(x_t, x_1) \, dx_1 \right\rangle dx_t$$

$$= \int \left\langle A(x_t), \int q(x_1) \, p_t(x_t \mid x_1) \, \nabla g(x_t) \, u_t(x_t \mid x_1) \, dx_1 \right\rangle dx_t$$

$$= \int \left\langle A(x_t), \nabla g(x_t) \int q(x_1) \, p_t(x_t \mid x_1) \, u_t(x_t \mid x_1) \, dx_1 \right\rangle dx_t$$

By definition, the marginal velocity field $v_t(x_t)$ is the expectation of the conditional field $u_t(x_t \mid x_1)$ over the posterior $p(x_1 \mid x_t) = \frac{q(x_1) p_t(x_t \mid x_1)}{p_t(x_t)}$. So, $v_t(x_t) = \int u_t(x_t \mid x_1) \frac{q(x_1) p_t(x_t \mid x_1)}{p_t(x_t)} \, dx_1$. Multiplying by $p_t(x_t)$ gives $p_t(x_t) v_t(x_t) = \int q(x_1) \, p_t(x_t \mid x_1) \, u_t(x_t \mid x_1) \, dx_1$. Substituting this back into our expression:

$$\ldots = \int \langle A(x_t), \nabla g(x_t) \, (p_t(x_t) v_t(x_t)) \rangle \, dx_t$$

$$= \int \langle A(x_t), p_t(x_t) B(x_t) \rangle \, dx_t$$

$$= \int p_t(x_t) \langle A(x_t), B(x_t) \rangle \, dx_t$$

This shows that the cross terms of $\mathcal{L}_{\text{cond}}$ and $\mathcal{L}_{\text{marg}}$ are also identical.

**(iii) Final Quadratic Term ($\| \cdot \|^2$):** The final term of $\mathcal{L}_{\text{cond}}$ is $\mathbb{E}_{x_1, x_t}[\|C(x_t, x_1)\|^2]$. We use the law of total variance: for a random variable $Z$, $\mathbb{E}[\|Z\|^2] = \|\mathbb{E}[Z]\|^2 + \text{Var}(Z)$. We apply this by first conditioning on $x_t$.

$$\mathbb{E}_{x_1, x_t}[\|C\|^2] = \mathbb{E}_{x_t \sim p_t} \left[ \mathbb{E}_{x_1 \sim p(x_1 \mid x_t)}[\|C(x_t, x_1)\|^2] \right]$$

$$= \mathbb{E}_{x_t} \left[ \|\mathbb{E}_{x_1 \mid x_t}[C(x_t, x_1)]\|^2 + \text{Var}_{x_1 \mid x_t}(C(x_t, x_1)) \right]$$

Let's compute the inner conditional expectation:

$$\mathbb{E}_{x_1 \mid x_t}[C(x_t, x_1)] = \mathbb{E}_{x_1 \mid x_t}[\nabla g(x_t) u_t(x_t \mid x_1)] = \nabla g(x_t) \mathbb{E}_{x_1 \mid x_t}[u_t(x_t \mid x_1)] = \nabla g(x_t) v_t(x_t) = B(x_t).$$

Substituting this back:

$$\mathbb{E}_{x_1, x_t}[\|C\|^2] = \mathbb{E}_{x_t} \left[ \|B(x_t)\|^2 + \text{Var}_{x_1 \mid x_t}(C(x_t, x_1)) \right]$$

$$= \mathbb{E}_{x_t}[\|B(x_t)\|^2] + \mathbb{E}_{x_t}[\text{Var}_{x_1 \mid x_t}(C(x_t, x_1))]$$

The first part, $\mathbb{E}_{x_t}[\|B(x_t)\|^2] = \int p_t(x_t)\|B(x_t)\|^2 \, dx_t$, is exactly the final term of $\mathcal{L}_{\text{marg}}$. The second part is the discrepancy term $\Delta(g)$:

$$
\begin{aligned}
\Delta(g) &= \mathbb{E}_{x_t}[\text{Var}_{x_1|x_t}(C(x_t, x_1))] \\
&= \mathbb{E}_{x_t}\left[\mathbb{E}_{x_1|x_t}\left[\|C(x_t, x_1) - \mathbb{E}_{x_1|x_t}[C(x_t, x_1)]\|^2\right]\right] \\
&= \mathbb{E}_{x_t}\left[\mathbb{E}_{x_1|x_t}\left[\|C(x_t, x_1) - B(x_t)\|^2\right]\right] \\
&= \mathbb{E}_{x_1, x_t}\left[\|C(x_t, x_1) - B(x_t)\|^2\right] \\
&= \iint q(x_1)\, p_t(x_t \mid x_1)\|\nabla g(x_t)u_t(x_t \mid x_1) - \nabla g(x_t)v_t(x_t)\|^2 \, dx_t\, dx_1 \\
&= \iint q(x_1)\, p_t(x_t \mid x_1)\|\nabla g(x_t)(u_t(x_t \mid x_1) - v_t(x_t))\|^2 \, dx_t\, dx_1
\end{aligned}
$$

**Conclusion:** Assembling all the terms, we have:

$$
\begin{aligned}
\mathcal{L}_{\text{cond}} &= \underbrace{\mathbb{E}_{x_t}[\|A\|^2]}_{\text{Term 1}} - \underbrace{2\mathbb{E}_{x_t}[\langle A, B\rangle]}_{\text{Term 2}} + \underbrace{(\mathbb{E}_{x_t}[\|B\|^2] + \Delta(g))}_{\text{Term 3}} \\
&= \left(\mathbb{E}_{x_t}[\|A\|^2] - 2\mathbb{E}_{x_t}[\langle A, B\rangle] + \mathbb{E}_{x_t}[\|B\|^2]\right) + \Delta(g) \\
&= \mathcal{L}_{\text{marg}} + \Delta(g)
\end{aligned}
$$

Since $\Delta(g)$ is the expectation of a squared norm, it is non-negative, which proves the theorem. $\square$

### A.4 PROOF OF PROPOSITION 2

*Proof.* The proof relies on the law of iterated expectation. Let $f(x_t)$ be any measurable function of $x_t$. The expectation of $f(x_t)$ over the marginal distribution $p_t(x_t)$ is:

$$
\mathbb{E}_{x_t \sim p_t}[f(x_t)] = \int_{\mathbb{R}^d} f(x_t) p_t(x_t) \, dx_t
$$

Now, we substitute the definition of the marginal path density, $p_t(x_t) = \int_{\mathbb{R}^d} q(x_1)p_t(x_t|x_1) \, dx_1$:

$$
\mathbb{E}_{x_t \sim p_t}[f(x_t)] = \int_{\mathbb{R}^d} f(x_t) \left(\int_{\mathbb{R}^d} q(x_1)p_t(x_t|x_1) \, dx_1\right) dx_t
$$

We can combine the terms inside a double integral:

$$
\mathbb{E}_{x_t \sim p_t}[f(x_t)] = \iint_{\mathbb{R}^d \times \mathbb{R}^d} f(x_t)q(x_1)p_t(x_t|x_1) \, dx_1 \, dx_t
$$

By Fubini's theorem, we can exchange the order of integration since the integrand is non-negative (or integrable):

$$
\mathbb{E}_{x_t \sim p_t}[f(x_t)] = \int_{\mathbb{R}^d} q(x_1) \left(\int_{\mathbb{R}^d} f(x_t)p_t(x_t|x_1) \, dx_t\right) dx_1
$$

This expression can be recognized as a nested expectation. The inner integral is the expectation of $f(x_t)$ over the conditional distribution $p_t(\cdot|x_1)$, and the outer integral is the expectation over the data distribution $q(x_1)$:

$$
\int_{\mathbb{R}^d} q(x_1) \left(\mathbb{E}_{x_t \sim p_t(\cdot|x_1)}[f(x_t)]\right) dx_1 = \mathbb{E}_{x_1 \sim q}\left[\mathbb{E}_{x_t \sim p_t(\cdot|x_1)}[f(x_t)]\right]
$$

$$
= \mathbb{E}_{x_1 \sim q, x_t \sim p_t(\cdot|x_1)}[f(x_t)]
$$

We have thus shown the general identity $\mathbb{E}_{x_t \sim p_t}[f(x_t)] = \mathbb{E}_{x_1 \sim q, x_t \sim p_t(\cdot|x_1)}[f(x_t)]$.

To prove the theorem, we simply choose $f(x_t)$ to be the squared residual of the marginal loss:

$$
f(x_t) = \left\|\mathcal{L}g(x_t) - \nabla_x g(x_t)\, v_t(x_t)\right\|^2
$$

By its definition, $\mathcal{L}_{marg} = \mathbb{E}_{x_t \sim p_t}[f(x_t)]$. Applying the identity we just derived gives:

$$
\mathcal{L}_{marg} = \mathbb{E}_{x_1 \sim q, x_t \sim p_t(\cdot|x_1)}\left[\left\|\mathcal{L}g(x_t) - \nabla_x g(x_t)\, v_t(x_t)\right\|^2\right]
$$

This completes the proof. $\square$

---

**Algorithm 1:** Koopman–CFM Training (simulation-free; fixed teacher, precomputed pairs)

---

**Input:** Fixed teacher velocity $v_{\text{CFM}}(t, x)$; encoder $g_\phi$; decoder $g_\psi^{-1}$; affine generator $\tilde{L}$; precomputed buffer
$\quad\quad \mathcal{B} = \{(x_0, x_1)\}$.
**Definition:** Lifted coordinate $\tilde{z}(t, x) := [\, 1, \, t, \, g_\phi(t, x)\,]^\top$.
**for** *each minibatch* **do**

$\quad\quad$ Sample $x_1 \sim q_1$, $t \sim \mathcal{U}(0, 1)$, then draw $x_t \sim p_t(\cdot \mid x_1)$;

$\quad\quad \mathcal{L}_{\text{cons}} \leftarrow \left\| \tilde{L}\, \tilde{z}(t, x_t) \, - \, Dg_\phi(t, x_t)[\, 1, \, v_{\text{CFM}}(t, x_t)\,] \right\|^2$;

$\quad\quad$ Sample $(x_0, x_1)$ from buffer $\mathcal{B}$;

$\quad\quad \mathcal{L}_{\text{phase}} \leftarrow \left\| \exp(\tilde{L})\, \tilde{z}(0, x_0) \, - \, \tilde{z}(1, x_1) \right\|^2$;

$\quad\quad \mathcal{L}_{\text{target}} \leftarrow \ell_{\text{img}}\!\Big( g_\psi^{-1}\!\big( \exp(\tilde{L})\, \tilde{z}(0, x_0)\big),\ x_1 \Big)$;

$\quad\quad \mathcal{L}_{\text{recon}} \leftarrow \ell_{\text{img}}\!\Big( g_\psi^{-1}\!\big( \tilde{z}(1, x_1)\big),\ x_1 \Big)$;

$\quad\quad \mathcal{L} \leftarrow \lambda_c \mathcal{L}_{\text{cons}} + \lambda_p \mathcal{L}_{\text{phase}} + \lambda_t \mathcal{L}_{\text{target}} + \lambda_r \mathcal{L}_{\text{recon}}$;
$\quad\quad$ Update $\{\phi, \psi, \tilde{L}\}$ by backprop on $\mathcal{L}$;

---

**Algorithm 2:** One-Step Koopman Sampling (matrix exponential + decode)

---

**Input:** Trained $(g_\phi, g_\psi^{-1}, L)$; prior $p_0 = \mathcal{N}(0, I)$.
**Input:** Lifted coordinate $z(t, x) := [\, 1, \, t, \, g_\phi(t, \, x)\,]^\top$.
Precompute $E \leftarrow \exp(L)$;
Sample $x_0 \sim p_0$;
**return** $\hat{x}_1 \leftarrow g_\psi^{-1}\!\Big( E\, z(0, \, x_0)\Big)$;

---

# B  EXPERIMENTAL DETAILS

This section complements Section 5 of the main paper by providing full details needed for reproducibility. We describe dataset prepration, model architecture and parametrization, training schedules, and computational resources.

## B.1  PARAMETERIZATION OF THE AFFINE LIFT

We parameterize $\tilde{L}$ with the following block structure.

$$\tilde{L} = \begin{bmatrix} 0 & 0 & \mathbf{0} \\ 1 & 0 & \mathbf{0} \\ \mathbf{b}_g & \mathbf{A}_{gt} & \mathbf{A}_{gg} \end{bmatrix} \tag{24}$$

This parameterization guarantees correct time evolution by design and yields affine dynamics for the observables: $\dot{g} = \mathbf{b}_g + \mathbf{A}_{gt} t + \mathbf{A}_{gg} g$. The learned parameters are the weights $\phi, \psi$ of the encoder $g_\phi$ and decoder $g_\psi^{-1}$ and the matrix blocks $(\mathbf{b}_g, \mathbf{A}_{gt}, \mathbf{A}_{gg})$.

**Data.**   We evaluate our approach on three datasets of increasing difficulty. MNIST contains 60,000 training and 10,000 test grayscale images of handwritten digits at resolution $28 \times 28$. FFHQ (Flickr-Faces-HQ) was downscaled to $32 \times 32$ resolution, from which we use all 70,000 RGB images. Finally, CIFAR-10 provides 50,000 training and 10,000 test images at resolution $32 \times 32$ across 10 object classes. This progression from simple digits to natural faces and general object classes allows us to systematically study the performance of our method as task complexity increases.

**Model Architecture.**   For all datasets, we employ a consistent backbone architecture: a SongUNet used as both encoder and decoder. To reduce the overall parameter count, we restrict the encoder output and decoder input to a single channel. Moreover, to obtain explicit control over the Koopman dimension, we optionally append a linear projection from the flattened UNet output to the target latent dimension.

**Training Details.** Before training our pipeline, we pre-trained an OT-CFM model following the reference implementation provided in the torchcfm code examples. From this model, we generated between $10^4$ and $10^6$ $(x_0, x_1)$ pairs (see Table 3 for exact counts per dataset), which served as inputs for computing the target loss. All models were trained using the Adam optimizer under identical training protocols across datasets. Experiments were carried out on NVIDIA A40, H100, and A100 GPUs. Additional hyperparameters, including learning rates, batch sizes, and training schedules, are reported in Table 3.

|  | **MNIST** | **FFHQ** | **CIFAR-10** |
|---|---|---|---|
| CFM iterations | 200k | 800k | 800k |
| Batch size | 128 | 256 | 124 |
| Learning rate | 0.0001 | 0.0001 | 0.0001 |
| Koopman iterations | 70k | 600k | 800k |
| Target weight (w/o $\mathcal{L}_{\text{cons}}$ – w/ $\mathcal{L}_{\text{cons}}$) | 1.0 – 1.0 | 1.0 – 0.01 | 1.0 – 0.01 |
| Operator Dimension | 1026 | 1026 | 1026 |
| UNet Output Channels | 1 | 1 | 1 |
| UNet Base Channels | 64 | 64 | 64 |
| UNet Channels Multiplier | [1,2,2] | [1,2,2,2] | [1,2,2,2] |
| Linear Projection | ✓ | ✗ | ✗ |

Table 3: Training hyperparameters for Koopman–CFM on MNIST, FFHQ, and CIFAR-10. *Linear projection* refers to the projection head at the UNet encoder output (resp. decoder input). Since loss terms are not of the same order of magnitude, the target loss was reweighted by the given parameter. *Koopman iterations* denote the number of iterations for the overall pipeline, while *CFM iterations* correspond to the underlying CFM model.

## C    ABLATIONS

This section expands the analysis of Section 5 by presenting ablations that clarify the role of each loss term, the effect of Koopman dimension, the impact of consistency on trajectory fidelity, and the interpretability of modes.

### C.1    IMPACT OF LOSS TERMS

Table 4 shows the effect of adding loss components across datasets. Phase and reconstruction alone yield poor FIDs, as they impose no constraint in image space. Adding the target loss improves fidelity by supervising decoded samples. Adding the consistency loss (weight 0.01) slightly worsens FID (e.g., FFHQ 7.5 → 10.1), since it regularizes the model to follow the teacher's nonlinear trajectories rather than shortcutting through straighter ones. This increases trajectory faithfulness at the cost of marginally higher endpoint error. We argue this tradeoff is beneficial: while endpoint-only distillation can optimize FID, it fails to capture the true generative flow (cf. Table 5, Fig. 8). Consistency-trained models achieve competitive FIDs while uniquely enabling spectral decomposition and robust downstream performance.

Table 4: Loss ablation across datasets showing the effect of incrementally adding loss components

| Dataset | $\mathcal{L}_{\text{recon}} + \mathcal{L}_{\text{phase}}$ | $\mathcal{L}_{\text{recon}} + \mathcal{L}_{\text{phase}} + \mathcal{L}_{\text{target}}$ | $\mathcal{L}_{\text{recon}} + \mathcal{L}_{\text{phase}} + 0.01\mathcal{L}_{\text{target}} + \mathcal{L}_{\text{cons}}$ |
|---|---|---|---|
| MNIST | 143.5 | 6.43 | 11.6 |
| FFHQ | 41 | 7.5 | 10.1 |
| CIFAR-10 | 64.5 | 16.7 | 14.1 |

Since we're optimizing a composite loss, there may be concerns of instability during training. For transparency we provide plots of the behavior of all our loss terms. Training is stable, and can be rationalized with well aligned objectives derived from both Koopman theory and the CFM framework.

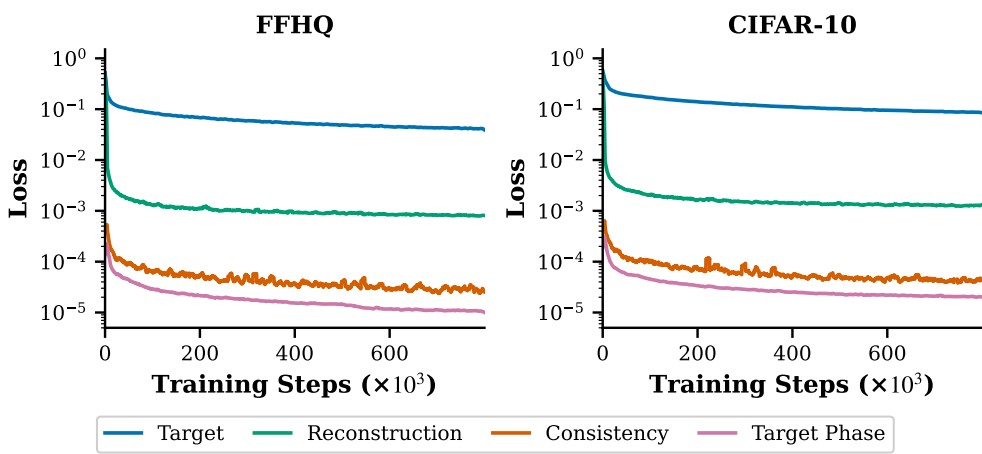

Figure 7: Training losses on CIFAR-10 and FFHQ datasets across different loss components.

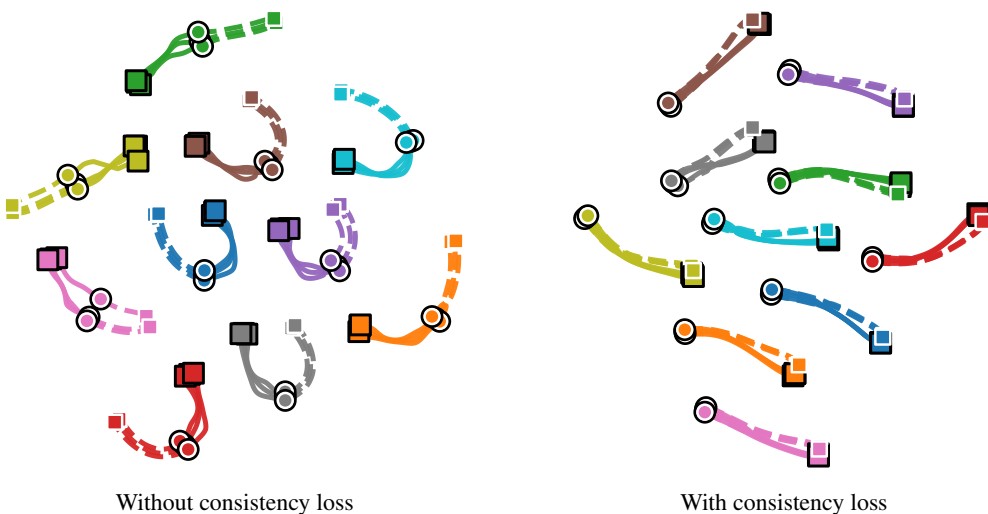

Without consistency loss                    With consistency loss

Figure 8: t-SNE visualization of CFM and Koopman trajectories in the embedding space on FFHQ. The consistency loss makes Koopman rollouts (dotted) follow the teacher dynamics (continuous) more closely. This is seen both in the proximity of trajectories and in the alignment of their endpoints. Circles mark starting points and squares mark end points.

## C.2 TRAJECTORY FIDELITY WITH AND WITHOUT CONSISTENCY LOSS

Table 5: MSE between trajectory rollouts between CFM and Koopman dynamics in latent space: we generate 1000 full trajectories $\{x_t\}$ via CFM encode in the Koopman latent space $g(t, x_t)$ and compare them with Koopman rollouts $g(x_t) = e^{Lt}g(t = 0, x_0)$.

| Dataset | Min | Max | Mean MSE | Std Dev |
|---|---|---|---|---|
| FFHQ (w/ consistency) | $3.0 \times 10^{-6}$ | $1.3 \times 10^{-5}$ | $5.0 \times 10^{-6}$ | $1.0 \times 10^{-6}$ |
| FFHQ (w/o consistency) | $5.24 \times 10^{-4}$ | $2.66 \times 10^{-3}$ | $1.30 \times 10^{-3}$ | $2.95 \times 10^{-4}$ |
| CIFAR-10 (w/ consistency) | $4.0 \times 10^{-6}$ | $3.7 \times 10^{-5}$ | $1.0 \times 10^{-5}$ | $4.0 \times 10^{-6}$ |
| CIFAR-10 (w/o consistency) | $3.46 \times 10^{-4}$ | $7.01 \times 10^{-3}$ | $1.74 \times 10^{-3}$ | $6.36 \times 10^{-4}$ |

We illustrate in Figure 8, the impact of consistency on trajectory fidelity. Notably, the consistency trained models trajectories closely tracks the teacher's nonlinear path. In contrast, the purely distilled

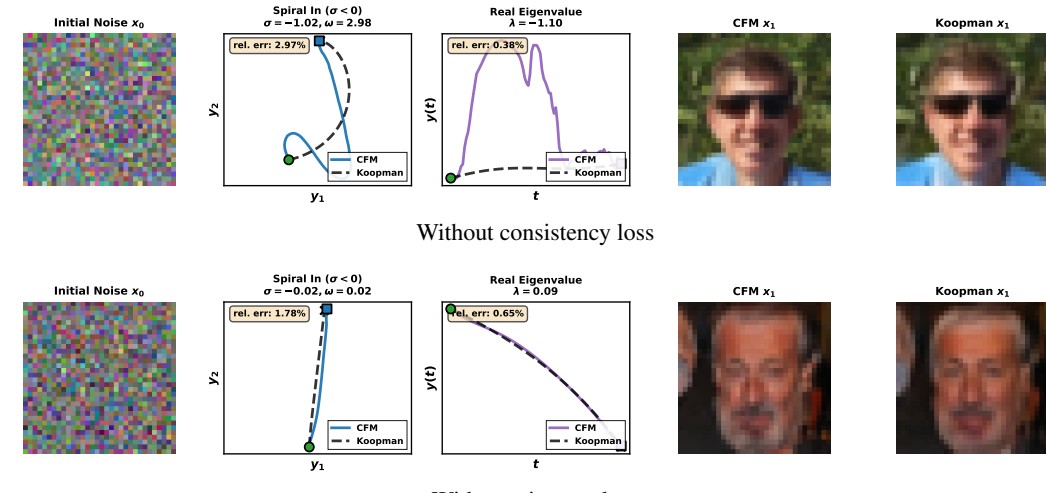

Figure 9: Trajectory comparison in Schur coordinates of the learned Koopman generator. With consistency, learned Koopman modes (dashed) accurately track CFM dynamics (solid). Without consistency, endpoints match but intermediate trajectories diverge, indicating the learned modes do not reflect true CFM dynamics.

model trajecotry diverges significantly, learning an unaligned shortcut, but with correct boundaries. This confirms that while a better FID can be achieved by ignoring the teacher's dynamics, doing so prevents the model from learning a faithful representation of the generative process.

To visualize trajectory fidelity in an interpretable coordinate system, we project dynamics onto the Schur basis of the learned generator $L$ (Figure 9). Each $2 \times 2$ block corresponds to a complex eigenvalue pair $\sigma \pm i\omega$, where $\sigma$ governs the exponential envelope and $\omega$ the oscillation frequency. With consistency training, the learned Koopman modes accurately track the CFM teacher's trajectory throughout, confirming that the representation captures the true generative dynamics. Without consistency, the endpoints remain correct, explaining the comparable generation quality, but the intermediate trajectory diverges significantly from the teacher. This demonstrates that consistency loss is essential for learning Koopman representations whose modes faithfully reflect the underlying flow, rather than merely learning a shortcut between boundaries.

## D  UNCURATED SAMPLES

This section supplements Section 5 by showing uncurated generations and reporting wall-clock sampling times, illustrating the tradeoffs between, speed, fidelity and interpretability.

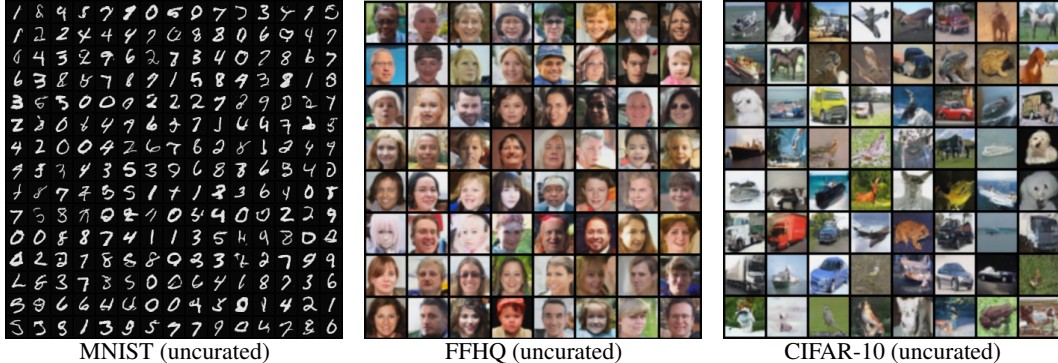

MNIST (uncurated)      FFHQ (uncurated)      CIFAR-10 (uncurated)

Figure 10: Uncurated samples from our Koopman generative model across three datasets. All samples are obtained via our one-step strategy.

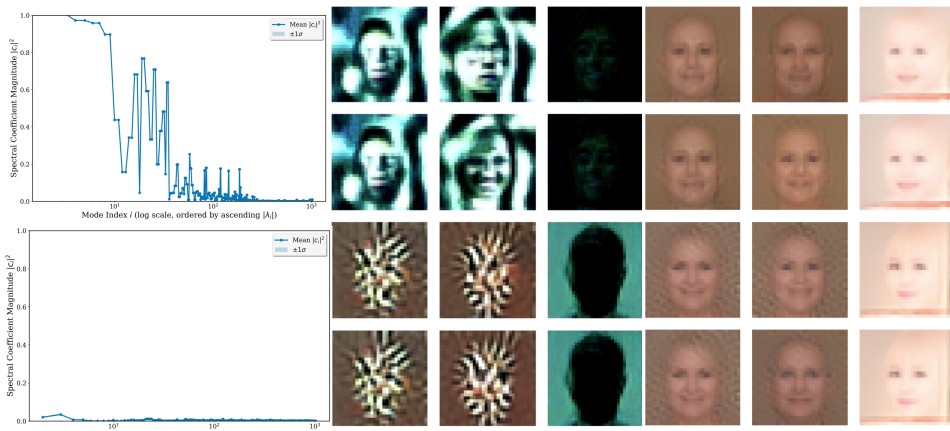

Figure 11: Left: Mean coefficients $|c_i|^2$ projected on the generator modes ordered by corresponding eigenvalue magnitude $|\lambda_i|^2$. Top corresponds to the spectrum along the modes obtained from training with consistency and bottom to those obtained from training without consistency Right: First three columns are some decoded modes of the generator trained with consistency loss, and the next three are those obtained from the generator trained without consistency.

## E  INTERPRETABILITY AND SPECTRAL ANALYSIS

### E.1  KOOPMAN MODE STRUCTURE

Figure 11 illustrates how consistency qualitatively changes the learned Koopman modes. Without consistency, individual modes tend to decode into entire faces—effectively full puzzle pieces—which suggests poor disentanglement, as each mode redundantly encodes the whole sample. By contrast, with consistency, the modes behave like localized "patch bases," decomposing faces into local interpretable components close to semantic components (e.g., hair, eyes). The spectral profile on the left of Fig. 11 also highlights important differences: with consistency, coefficients decay with eigenvalue magnitude, whereas without consistency the spectrum remains flat, indicating the absence of structured decomposition.

### E.2  INSIGHTS ON TEACHER TRAINING

Similarity matrices (Fig. 15, left) reveal a clear ordering: when modes are sorted by $\mathrm{Re}(\lambda)$, mid-training checkpoints already align with the low-decay modes of the final model, while early checkpoints show little correspondence. This is further quantified by the cumulative similarity (right), which increases monotonically across training stages.

### E.3  GENERATION PROCESS: COARSE-TO-FINE.

To investigate the interpretability of the learned Koopman representation, we perform progressive mode reconstruction by truncating the eigenspectrum of the generator $L$. Specifically, we compute the eigendecomposition of the feature block $A_{gg} = L_{[2:,2:]}$ and construct a real-valued basis by taking $\mathrm{Re}(v)$ and $\mathrm{Im}(v)$ for each complex conjugate eigenvector pair. We sort modes by the real part of their eigenvalues, $\mathrm{Re}(\lambda)$, which governs the exponential timescale of each mode: more negative values correspond to strongly decaying dynamics while values closer to zero or positive correspond to slowly decaying or amplifying dynamics.

Given an encoded image $z = [1, t, g]^\top$ where $g \in \mathbb{R}^{1024}$ denotes the feature vector, we reconstruct using only the first $k$ modes by projecting onto the truncated basis $B_k \in \mathbb{R}^{1024 \times k}$:

$$\hat{g}_k = B_k B_k^\dagger g, \tag{25}$$

where $B_k^\dagger$ denotes the pseudoinverse. The reconstructed features $\hat{g}_k$ are then decoded back to image space.

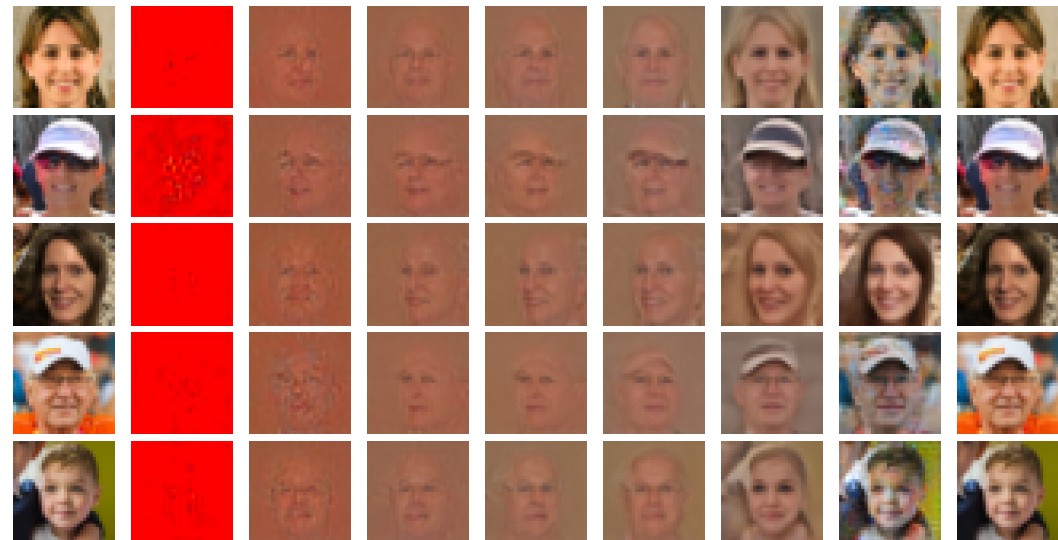

Figure 12: Progressive mode reconstruction sorted by $\text{Re}(\lambda)$. Slow modes (negative $\text{Re}(\lambda)$) capture coarse structure, while fast modes (positive $\text{Re}(\lambda)$) add fine details. The learned Koopman spectrum provides an interpretable hierarchy reflecting the multi-scale nature of the generative process.

Figure 12 shows reconstructions for increasing $k$. The slowest modes ($k \leq 200$, $\text{Re}(\lambda) \leq -0.58$) produce homogenous outputs, indicating these modes encode a global bias that requires additional modes to balance. At $k = 400$, coarse facial structure emerges, face shape, average skin tone, and approximate feature positions. As $k$ increases to 600–800, identity-specific features begin to appear, though images remain soft. Finally, modes with $\text{Re}(\lambda) > 0$ ($k > 1000$) contribute fine details: hair texture, sharp edges, and accessories such as hats and glasses. Full reconstruction recovers the original image with high fidelity.

These results demonstrate that the Koopman eigenspectrum induces a principled coarse-to-fine hierarchy: slow modes capture global structure while fast modes encode high-frequency details. Unlike PCA, which orders components by variance, this ordering emerges from the *dynamics* of the generative flow, providing an interpretable decomposition tied to the underlying generative process.

### E.4    KOOPMAN MODES ALIGN WITH SEMANTIC DIRECTIONS

We provide uncurated qualitative examples of Koopman mode-induced image edits in Figure 13.a, Figure 13.c, Figure 13.d, Figure 13.b. Each row shows a different test image, with columns corresponding to perturbation strengths $\alpha \in \{-0.2, -0.1, 0, 0.1, 0.2\}$. Importantly, these modes were not manually selected; rather, they were automatically identified by ranking all eigenmodes according to their CLIP coherence scores with respect to each attribute prompt.

**Sunglasses (Mode 1019).**    For the model trained with consistency loss, this mode demonstrates strong semantic alignment: positive $\alpha$ consistently introduces sunglasses across diverse subjects while preserving identity, pose, and background. Negative $\alpha$ produces the inverse effect, brightening the eye region and removing eyewear. The transformation generalizes across ages, genders, and lighting conditions, confirming that this eigenmode captures a disentangled semantic direction rather than spurious correlations.

**Brown Hair (Mode 767).**    This mode exhibits coupling between hair color and global illumination. While positive $\alpha$ shifts toward darker hair tones, it simultaneously reduces overall image brightness. This entanglement suggests that some semantic attributes share spectral structure in the Koopman operator, consistent with the lower selectivity scores reported in Table 4.

**Effect of Consistency Loss.**    Without consistency loss, perturbing Koopman modes produces no discernible change in the decoded images, regardless of the perturbation magnitude $\alpha$. In contrast,

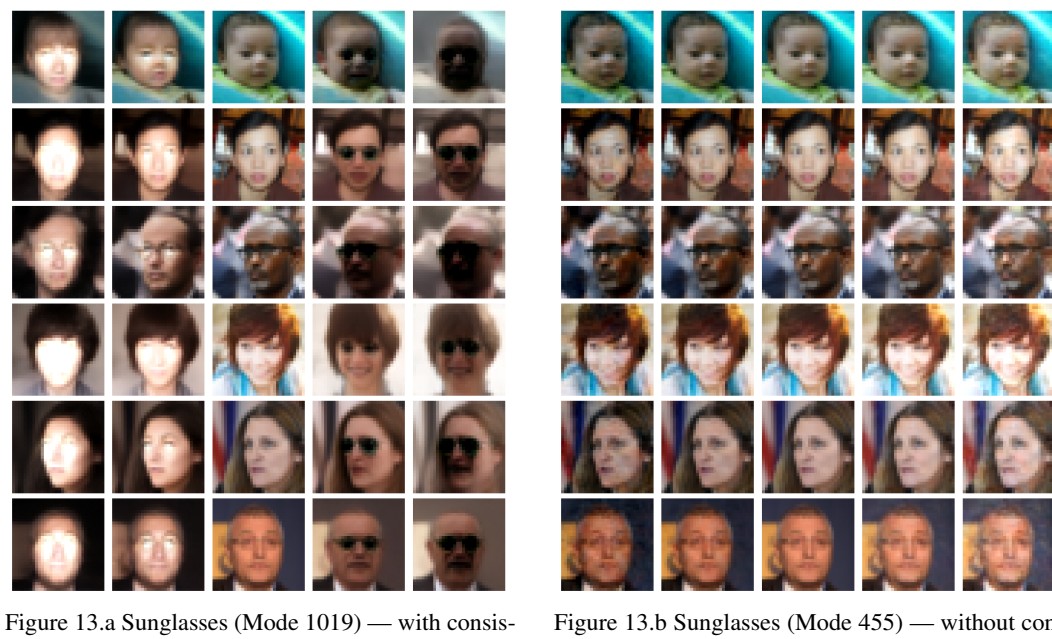

Figure 13.a Sunglasses (Mode 1019) — with consistency

Figure 13.b Sunglasses (Mode 455) — without consistency

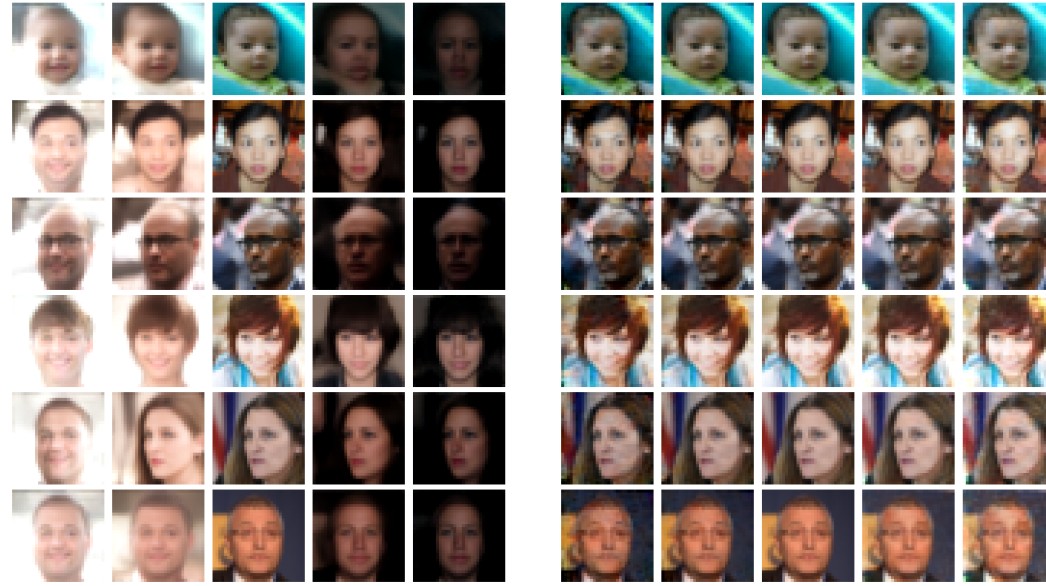

Figure 13.c Brown hair (Mode 767) — with consistency

Figure 13.d Brown hair (Mode 2) — without consistency

Figure 13: **Koopman mode perturbations for semantic editing.** Each grid shows different test subjects (rows) perturbed with $\alpha \in \{-0.2, -0.1, 0, 0.1, 0.2\}$ (columns). Modes were automatically identified via CLIP coherence analysis. **(a, c)** With consistency loss, perturbing individual eigenmodes produces semantically meaningful edits—adding sunglasses or darkening hair—while preserving identity. **(b, d)** Without consistency loss, the same perturbations yield no visible change, demonstrating that consistency training is essential for learning actionable semantic directions.

the consistency-trained model yields clearly visible and semantically coherent edits. This qualitative difference corroborates the quantitative findings in Section 5.4: consistency loss is essential not only for learning a faithful linear decomposition of the dynamics, but also for ensuring that the resulting eigenmodes correspond to actionable semantic directions in image space.

| | No Consistency | | Consistency | |
|---|---|---|---|---|
| Direction | CLIP ↑ | LPIPS ↓ | CLIP ↑ | LPIPS ↓ |
| *Direct Koopman Editing* | | | | |
| Hat | 0.069 | **0.018** | **0.077** | 0.019 |
| Sunglasses | 0.070 | 0.035 | **0.072** | **0.034** |
| Smile | **0.093** | **0.016** | 0.090 | 0.020 |
| Age | 0.121 | **0.035** | **0.123** | 0.036 |
| Gender (M→W) | 0.153 | **0.021** | **0.157** | 0.025 |
| Gender (W→M) | 0.150 | **0.031** | **0.152** | 0.035 |
| *CFM-Based Editing* | | | | |
| Hat | **0.148** | 0.060 | 0.094 | **0.053** |
| Sunglasses | **0.130** | 0.076 | 0.107 | **0.056** |
| Smile | **0.087** | 0.013 | 0.052 | **0.010** |
| Age | **0.186** | 0.097 | 0.145 | **0.087** |
| Gender | **0.178** | 0.091 | 0.139 | **0.043** |

Table 6: Quantitative comparison of latent directions, at $\alpha = 3.0$

# F    APPLICATIONS

## F.1    INVERSION

To edit a real image $\mathbf{x}$, we first invert it to a corresponding noise sample $\mathbf{x}_0$ such that integrating the CFM ODE recovers the original image. We encode the image to its lifted representation $\mathbf{z} = g(\mathbf{x})$ and compute the target noise-space embedding $\mathbf{z}_0^* = \exp(-L)\mathbf{z}_1$. We then optimize $\mathbf{x}_0$ to match this target:

$$\mathbf{x}_0^* = \arg\min_{\mathbf{x}_0} \|g(\mathbf{x}_0) - \mathbf{z}_0^*\|_2^2 \tag{26}$$

We show inversion examples in Figure 14. We again highlight the difference with the no-consistency model, which introduces artifacts and have noticeable unstable optimization.

## F.2    DISCOVERING SEMANTIC DIRECTIONS IN KOOPMAN LATENT SPACE

We discover semantic directions in the lifted Koopman space using a supervised approach. Given the FFHQ dataset, we first encode each image $\mathbf{x}$ into its lifted representation

$$\mathbf{z} = g(\mathbf{x})$$

at $t = 1$. Binary attribute labels (e.g., smiling vs. not smiling, eyeglasses vs. no eyeglasses) are obtained via CLIP classification using natural-language prompts.

For each binary attribute, we compute a semantic direction as the difference between class-conditional mean embeddings:

$$\mathbf{d}_{\text{attr}} = \mathbb{E}[\mathbf{z} \mid y = 1] - \mathbb{E}[\mathbf{z} \mid y = 0]. \tag{27}$$

Semantic editing is performed via linear traversal in the Koopman latent space:

$$\mathbf{z}_{\text{edited}} = \mathbf{z} + \alpha\,\mathbf{d}_{\text{attr}}, \tag{28}$$

where $\alpha$ controls the edit strength. The edited latent code is then decoded back to image space via

$$\hat{\mathbf{x}} = g^{-1}(\mathbf{z}_{\text{edited}}).$$

We show, in Table 6, that the provided latent directions are better with the consistency model, both in Koopman or unlifted to the image space, with CLIP and LPIPS evaluations for different attributes.

### F.3 Noise engineering: Performing image editing by optimizing CFM noise perturbation

A key advantage of our Koopman-based framework is that semantic directions discovered in the lifted space can be transferred to perform editing with the original CFM model. This demonstrates that the Koopman operator captures meaningful structure that generalizes beyond the learned decoder.

**Optimizing semantic perturbations.** Rather than directly adding $\mathbf{d}_{\text{attr}}^{(0)}$ in pixel space, we optimize a perturbation $\Delta\mathbf{x}_0$ such that the perturbed noise induces the desired semantic shift in the lifted space:

$$\Delta\mathbf{x}_0^* = \arg\min_{\Delta\mathbf{x}_0} \left\| g(\mathbf{x}_0 + \Delta\mathbf{x}_0) - \left( g(\mathbf{x}_0) + \mathbf{d}_{\text{attr}}^{(0)} \right) \right\|_2^2 \tag{29}$$

Edited images are then generated by integrating the perturbed noise through the *original* CFM model:

$$\hat{\mathbf{x}}_1 = \int_0^1 v_\theta(t, \mathbf{x}_0 + \alpha\,\Delta\mathbf{x}_0^*)\, dt \tag{30}$$

where $\alpha$ controls the edit strength and $v_\theta$ is the pretrained CFM velocity field.

**Role of consistency regularization.** We observe a stark difference in editing quality depending on whether the Koopman model was trained with consistency loss. Figure 16 and Figure 17 compares semantic traversals for models trained with and without this loss term.

With consistency regularization, the optimized perturbations remain well-behaved across a wide range of edit strengths ($\alpha \in [0, 3]$). Edits are semantically meaningful, identity is preserved, and image quality remains stable even at large $\alpha$ values. In contrast, without consistency regularization, edited images exhibit severe degradation at moderate-to-large perturbation strengths: backgrounds become corrupted with color artifacts, facial structure deteriorates, and identity is lost.

We attribute this to the role of consistency loss in aligning the Koopman dynamics with the underlying CFM trajectory. When this alignment is enforced, the learned operator $\exp(L)$ accurately models how features evolve under the flow, ensuring that mapped directions $\mathbf{d}_{\text{attr}}^{(0)} = \mathbf{d}_{\text{attr}}\exp(-L)$ correspond to valid perturbations within the noise distribution's support. Without this constraint, the backward mapping may produce directions that push samples off the data manifold, causing the CFM integration to generate out-of-distribution outputs.

These results highlight that our Koopman framework not only enables direct editing via the learned decoder $g^{-1}$, but also provides a principled mechanism for *noise engineering*, transferring semantic control to any compatible generative model by operating in its noise space.

### F.4 Functional robustness on downstream tasks

Finally, we evaluate if this interpretable structure of our framework translates to challenging downstream tasks: inpainting, super-resolution, and denoising. These tasks test the model's ability to perform conditional generation, which depends on the quality of its learned dynamics. For a corrupted input encoded to $z_{1,corr}$, we reconstruct by adding noise at $t = 0$ and evolving it through the learned process:

$$z_{0,corr} = e^{-L} z_{1,corr} \quad ; \quad x_{recon} = g_\psi^{-1}(e^L(z_{0,corr} + \text{noise}))$$

As shown in Figure 18, the consistency-trained model significantly outperforms the ablation model across all tasks. This superior performance is a direct consequence of the structured, Fourier-like basis described above. Because its learned dynamics can induce local, patch-based semantic modifications, the model is uniquely equipped to solve tasks that require local reasoning, like inpainting a missing patch. The purely distilled model fails and simply reproduces the same image, showing that it only learned the noise-to-data map, instead of the underlying image data distribution.

## G Extended survey on interpretability of generative models

There is a rich body of work on understanding how generative models transform noise into data. Early research on VAEs and GANs focused on analyzing their latent spaces. Variational Autoencoders were

used to learn *disentangled* representations of data Bengio et al. (2013), i.e., latent codes that separate the underlying generative factors of variation Higgins et al. (2016); Burgess et al. (2018); Kim & Mnih (2018); Khemakhem et al. (2020). The success of Generative Adversarial Networks Goodfellow et al. (2014) prompted similar studies Chen et al. (2016). Because the latent space of GANs is not explicitly structured, research focused on identifying directions that correspond to interpretable generative factors, enabling controlled image editing Jahanian et al. (2020); Härkönen et al. (2020); Voynov & Babenko (2020); Shen & Zhou (2021). The rise of diffusion and flow models as state-of-the-art generative methods naturally raised the question of whether such interpretability techniques could be extended to these models. However, their iterative generation process and the prevalence of complex, learnable control mechanisms Zhang et al. (2023) have not yielded equally simple or powerful methods for interpretation and editing. Existing approaches tend to be more complicated and lack the conceptual clarity and usability of those developed for VAEs and GANs Kwon et al. (2022); Yang et al. (2023); Meng et al. (2022); Kulikov et al. (2024). In contrast, our method preserves the dynamical-systems view of these models while enabling simple and interpretable latent-space manipulations.

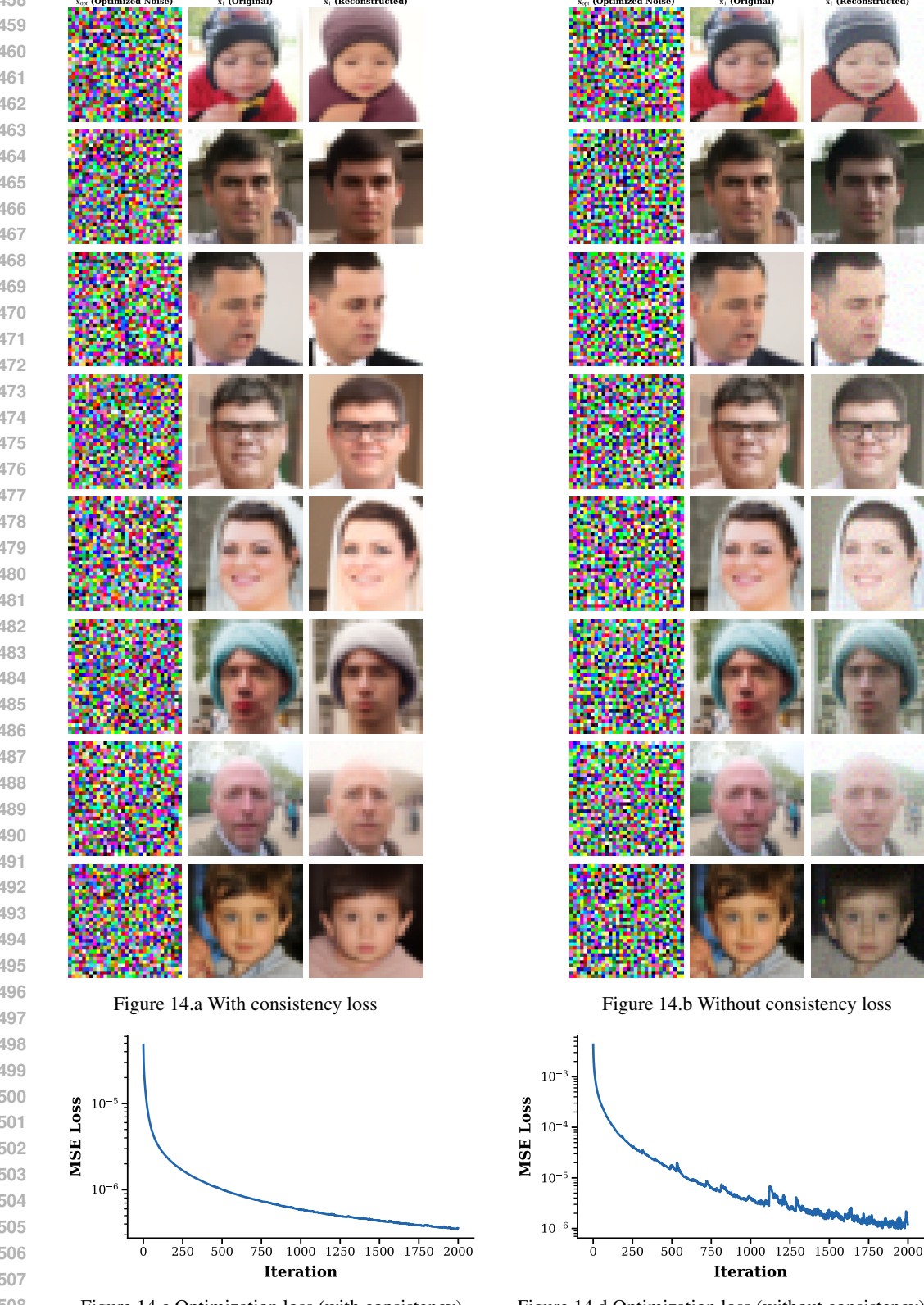

Figure 14.a With consistency loss

Figure 14.b Without consistency loss

Figure 14.c Optimization loss (with consistency)

Figure 14.d Optimization loss (without consistency)

Figure 14: **CFM inversion via backward Koopman evolution.** Each row in (a, b) shows: optimized noise $x_{\text{opt}}$, original image $x_1$, and CFM reconstruction $\hat{x}_1$. **(a)** With consistency loss, reconstructions faithfully preserve identity. **(b)** Without consistency, reconstructions appear plausible but the inversion is not principled. **(c, d)** Optimization loss curves reveal that consistency loss yields smooth convergence, while without it the landscape is ill-conditioned with higher final loss.

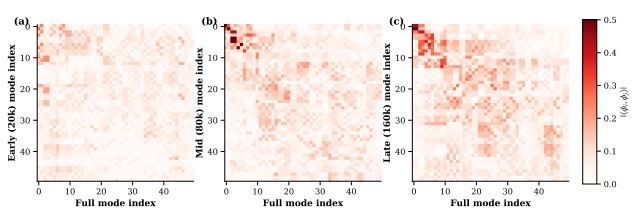

Figure 15.a Eigenmode similarity matrices comparing early and mid-training checkpoints against the fully trained model.

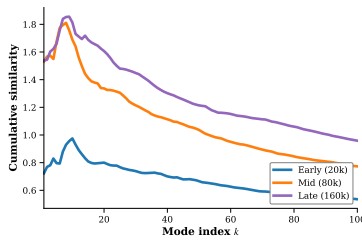

Figure 15.b Cumulative average diagonal similarity showing progressive mode acquisition during training.

Figure 15: Koopman mode acquisition during training. (Left) Eigenmode similarity matrices sorted by $\mathrm{Re}(\lambda)$; diagonal structure at mid-training indicates decaying modes are learned first. (Right) Cumulative similarity confirms mid-training acquires low real part modes versus minimal correspondence at early training.

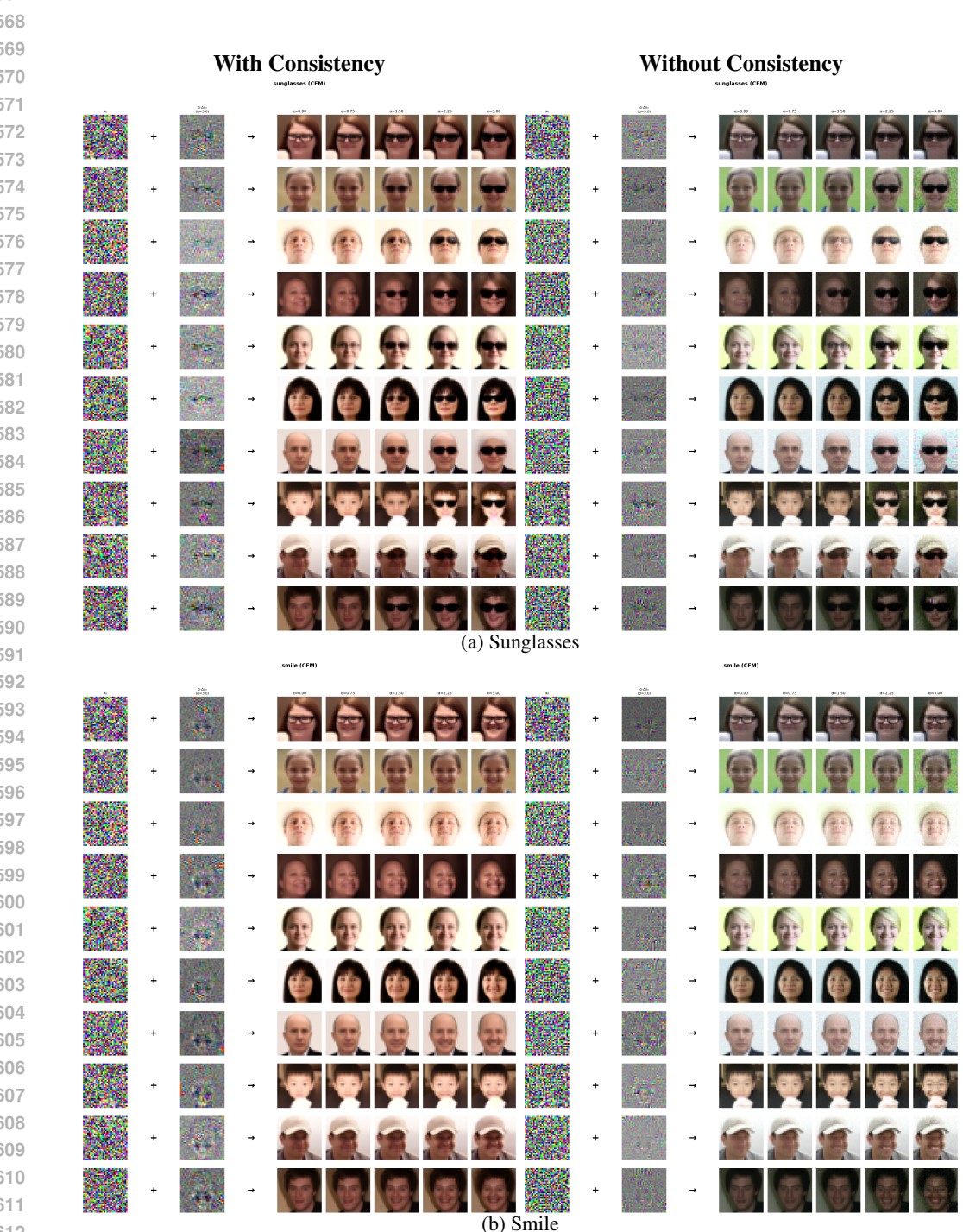

Figure 16: **Consistency loss enables stable semantic editing.** Edits via optimized noise perturbations $\mathbf{x}_0 + \alpha\Delta\mathbf{x}_0$ integrated through CFM ($\alpha \in [0, 3]$, increasing left-to-right). With consistency loss (left), edits remain coherent and identity-preserving. Without (right), large $\alpha$ causes artifacts and structural collapse.

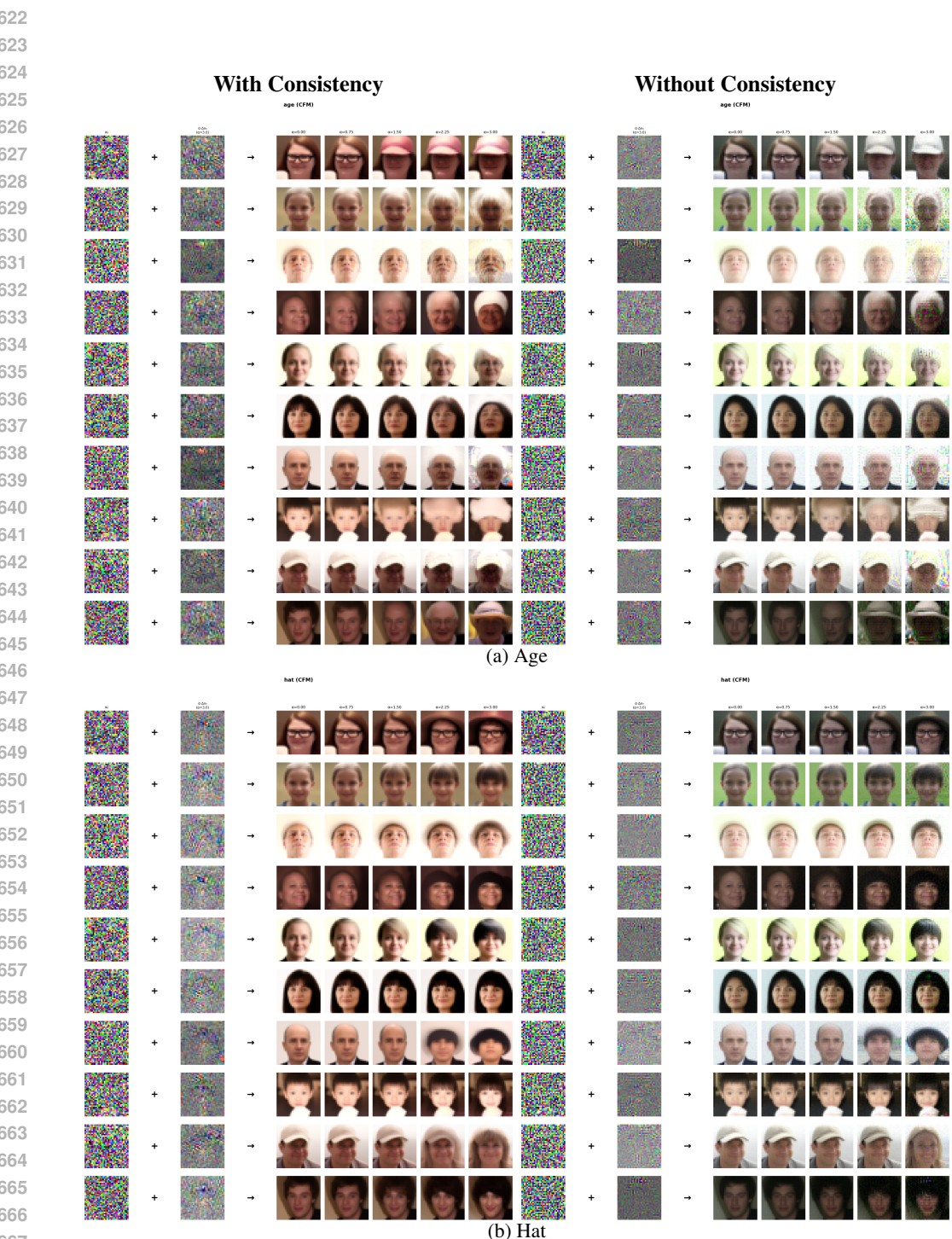

Figure 17: **Additional semantic directions.** Same setup as Figure 16. Consistency loss enables stable traversal across diverse attributes.

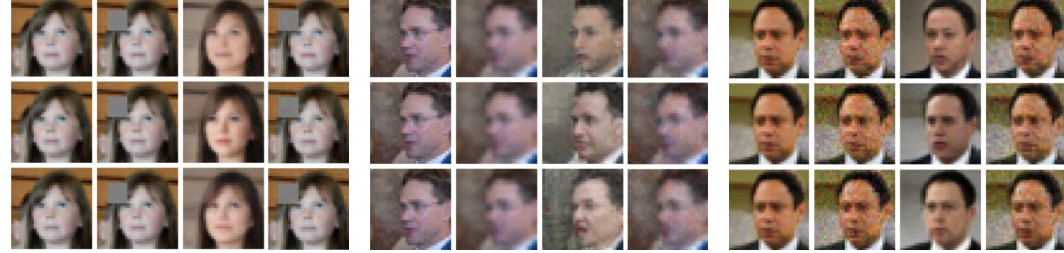

Figure 18.a Inpainting       Figure 18.b Super-Resolution       Figure 18.c Denoising

Figure 18: Performance on structured generative tasks. For each task, we show the input, the corrupted image, the result from our consistency-trained model, and the result from the ablation model. Each row corresponds to the application of different gaussian noise. Our model consistently produces coherent, high-fidelity results, while the ablation model fails.

