# OpenReview forum: "Unfolding Generative Flows"
_ICLR.cc/2026/Conference — ICLR 2026 Conference Desk Rejected Submission_

### Official Review · Reviewer_mSkg · 2025-10-25

**Soundness:** 2
**Presentation:** 2
**Contribution:** 2
**Rating:** 4
**Confidence:** 3

**Summary:**

This paper proposes a novel acceleration and interpretation framework for Continuous Normalizing Flows (CNFs) using Koopman operator theory. By lifting nonlinear CNF dynamics into a linear latent space, the model enables one-step sampling via a matrix exponential and offers spectral interpretability through Koopman eigenmodes. A simulation-free consistency loss ensures fidelity to the teacher CNF’s vector field.

**Strengths:**

1. This paper presents an algebraic perspective on generative flows by applying Koopman operator theory to Continuous Normalizing Flows (CNFs).

2. The Koopman eigenmodes uncover semantically meaningful latent directions, enabling controllable and interpretable manipulations of generated samples.

3. The proposed method replaces multi-step ODE integration with a single matrix exponential, yielding significant speedups while preserving sample quality.

**Weaknesses:**

1. The experimental evaluation includes limited baselines, making it difficult to fully assess the significance and comparative advantages of the proposed method. The results in Table 1 do not clearly highlight its superiority over existing approaches.

2. Computing large matrix exponentials may become computationally expensive or numerically unstable when applied to high-dimensional or high-resolution data.

3. The method depends on a pre-trained CNF as a teacher, resulting in a two-stage distillation process rather than a fully end-to-end training framework.

**Questions:**

1. The training objective involves four loss coefficients, but the paper does not clearly explain how these terms are balanced or tuned during optimization.

2. The Koopman model without the consistency loss reportedly achieves better performance than the version with it. This raises the question of the practical role and necessity of the consistency loss in the overall framework.

3. Since the matrix exponential is defined as an infinite series, it is unclear how it is approximated in practice. Moreover, for large matrices
L, computing the exponential can be computationally expensive. How to reduce this cost or improve numerical efficiency?

---

> ### Author Response · Authors · 2025-11-21
>
> > The experimental evaluation includes limited baselines, making it difficult to fully assess the significance and comparative advantages of the proposed method. The results in Table 1 do not clearly highlight its superiority over existing approaches.
>
> We acknowledge the reviewer's comment. We plan to add relevant baselines, namely, MeanFlow, Consistency Flow, and Simulation Free Matching, as suggested by EtF6, in the final version. Also mentioned in the general response above, thanks to the decomposition of the learned Koopman operator, our method provides unprecedented insights and interpretability of the CFM dynamics, and is more than a one-step generation contribution. In this context, relying only on the generation quality (FID) for evaluation doesn't fully encompass the full potential of our method.
>
> > Computing large matrix exponentials may become computationally expensive or numerically unstable when applied to high-dimensional or high-resolution data.
>
> We thank the reviewer for the relevant inquiry. However, we would like to point out that the biggest matrix in our experiments are 1024x1024, a common matrix dimension for modern GPUs. We used Pytorch's ```matrix_exp``` GPU implementation (according to the current implementation, it is based on the peer-reviewed paper [1], which shows optimal approximation and speed) across all experiments, and did not see any particular instabilities in the training losses. We plan to include loss curves of all the terms involved in the composite training objective to highlight the stable behavior.
>
> > The method depends on a pre-trained CNF as a teacher, resulting in a two-stage distillation process rather than a fully end-to-end training framework.
>
> We agree with the reviewer that a fully end-to-end training framework would be great! However, as far as we know, most distillation methods require at least a coarse pretrained network to distill the teacher correctly, relying on two-stage approaches to provide the best possible result. We left this improvement for future work.
>
> > The training objective involves four loss coefficients, but the paper does not clearly explain how these terms are balanced or tuned during optimization.
>
> We provide in Appendix 3 training details, including training loss weights. Notably, we weigh the losses so that they contribute to the final loss with a similar magnitude. The loss weights remain fixed during training:
> 1. Phase loss (Eq. 6): 1.0
> 2. Target loss (Eq. 7): 0.01
> 3. Reconstruction loss (Eq. 8): 1.0
> 4. Consistency loss: 1.0
>
> > The Koopman model without the consistency loss reportedly achieves better performance than the version with it. This raises the question of the practical role and necessity of the consistency loss in the overall framework.
>
> Thanks for the relevant question. As mentioned in the general response above, thanks to the decomposition of the learned Koopman operator, our method provides unprecedented insights and interpretability of the CFM dynamics, and is more than a one-step generation contribution. In this context, relying only on the generation quality (FID) for evaluation doesn't fully encompass the full potential of our method. We plan to improve the interpretability section presentation and to complete it with a quantitative evaluation of the semantic coherence of Koopman modes.
>
> > Since the matrix exponential is defined as an infinite series, it is unclear how it is approximated in practice. Moreover, for large matrices L, computing the exponential can be computationally expensive. How to reduce this cost or improve numerical efficiency?
>
> As said above (see also answer to Reviewer mSkg), we used Pytorch's ```matrix_exp``` GPU implementation, based on [1], across all experiments. Also, the biggest matrices exponentiated in our experiments are 1024x1024, a common matrix dimension for modern GPUs. We will provide compute-time of the different operations inside the Koopman module in the rebuttal.
>
> As said above, if you think that we missed an important point in our response and improvement plan, we will be happy to address it during the discussion.
>
> [1] "Bader, P.; Blanes, S.; Casas, F. "Computing the Matrix Exponential with an Optimized Taylor Polynomial Approximation." Mathematics 2019

---

> ### Author Response · Authors · 2025-12-03
>
> >The experimental evaluation includes limited baselines, making it difficult to fully assess the significance and comparative advantages of the proposed method. The results in Table 1 do not clearly highlight its superiority over existing approaches.
>
> As requested, we now compare against strong one-step samplers, including Rectified Flow and MeanFlow. This addresses concerns that our method’s performance was not positioned clearly relative to SOTA. Again, we highlight that the main appeal of our method is not the generation quality, but the consistency with the teacher and the interpretability, possible with the Koopman operator decomposition.
>
> > Computing large matrix exponentials may become computationally expensive or numerically unstable when applied to high-dimensional or high-resolution data.
>
> Scaling to dimension 64x64 empirically - with fixed Koopman dimension validates the stability of the exponentiation! We also added loss curves (Appendix C) to shed light on the training stability. Despite operating with a 1024x1024 matrix exponentiation, the optimization process behaves smoothly.
>
> > The training objective involves four loss coefficients, but the paper does not clearly explain how these terms are balanced or tuned during optimization.
>
> We added loss curves in Appendix C to illustrate the optimization stability.
>
> > The Koopman model without the consistency loss reportedly achieves better performance than the version with it. This raises the question of the practical role and necessity of the consistency loss in the overall framework.
>
> The revised paper clarifies why the consistency loss is structurally necessary even though FID is slightly lower without it. As shown in Table 2 and Figs. 3–6, removing consistency collapses the Koopman representation into an endpoint-only distillation: trajectories no longer match the teacher (260× higher error), eigenmodes lose semantic meaning, and spectral structure becomes meaningless. In contrast, the consistency-trained version maintains faithful dynamics, enabling the new applications added in the revision—GAN-like semantic directions, inversion via a teacher aligned operator exp(−L), and porting edits back to the original CFM (Fig. 7). Thus, while endpoint distillation may optimize FID marginally, it destroys the linearization that defines our method; the consistency loss is essential for achieving a valid Koopman operator and the interpretability/editability capabilities highlighted in the revision.

---

### Official Review · Reviewer_5DZh · 2025-10-27

**Soundness:** 2
**Presentation:** 3
**Contribution:** 3
**Rating:** 4
**Confidence:** 3

**Summary:**

The paper proposes a Koopman-operator view of Conditional Flow Matching (CFM). It proposes to distill an encoder, a decoder and a generator matrix from an existing CFM teacher to connect state space to the Koopman representation so that the non-linear CFM dynamics become linear and thus analytically solvable in one step by a matrix exponential. A key technical piece is a simulation-free infinitesimal consistency loss that aligns the learned observables with the teacher vector field along the entire path. Empirically, on MNIST, CIFAR-10, and 32×32 FFHQ with an OT-CFM teacher, the one-step sampler yields competitive FIDs and large speedups, and exposes interpretable spectral identities.

**Strengths:**

- Distilling CFM dynamics into a globally linear dynamics enabling 1-step analytical sampling and spectral analysis is interesting. The time-augmentation + affine-lift construction is clean. Has wall-clock improvements compared to multi-NFE CFM.
- Ablation showing that removing the consistency term can improve FID but harms trajectory fidelity. This justifies the method’s design.
- The paper is written clearly and describes the necessary prerequisits for readers.

**Weaknesses:**

- The paper itself notes challenges scaling to high-resolution images. Exponentiating a large dense matrix can be costly and numerically delicate. Results stop at 32×32.
- Baselines focus on OT-CFM and Consistency Flow Matching. Missing broader experiment settings on other variants of CFM models or comparisons to SOTA few/one-step approaches.
- Paper claims a key contribution of interpretability by exposing the Koopman dynamics. Two methods of interpretation are discussed: Dirac delta perturbations and spectral decomposition. However, it's not intuitively clear how such interpretations could shed more insight into flow matching training. Also not clear about the implications of image effects after perturbation in Figure 3.

**Questions:**

1. Beyond figures in the paper, can authors make the interpretability claim more rigorous or discuss more about potential implication of such interpretation on training?
2. How does performance vary with teacher quality? If the teacher is changed, does the learned $L$ differ qualitatively in interpretations (e.g., spectrum)?
3. Do eigen-modes learned on one dataset transfer to others? Any mode-sharing across classes in CIFAR-10?

---

> ### Author Response · Authors · 2025-11-21
>
> > The paper itself notes challenges scaling to high-resolution images. Results stop at 32×32.
>
> We acknowledge the reviewer's concern. However, we would like to emphasize that the complexity scales from MNIST $\to$ FFHQ $\to$ CIFAR. We plan to add experiments on $64\times64$ FFHQ to show that our method also scales in resolution.
>
> > Exponentiating a large dense matrix can be costly and numerically delicate.
>
> Indeed, exponentiation can be numerically delicate. However, we would like to emphasize that the matrix L is trained over 100000 steps, and that we noticed no significant instability affecting the training. For the implementation, we used Pytorch's  ```matrix_exp``` GPU implementation across all experiments (according to the current implementation, it is based on the peer-reviewed paper [1], which shows optimal approximation and speed). We plan to include loss curves of all the terms involved in the composite training objective to highlight the stable behavior.
>
> > Baselines focus on OT-CFM and Consistency Flow Matching. Missing broader experiment settings on other variants of CFM models or comparisons to SOTA few/one-step approaches.
>
> We acknowledge the reviewer's comment. We plan to add relevant baselines, namely, MeanFlow, Consistency Flow and Simulation Free Matching, as suggested by EtF6 and 12ML, in the final version.
>
> > Paper claims a key contribution of interpretability by exposing the Koopman dynamics. Two methods of interpretation are discussed: Dirac delta perturbations and spectral decomposition. However, it's not intuitively clear how such interpretations could shed more insight into flow matching training. Also not clear about the implications of image effects after perturbation in Figure 3.
>
> We thank the reviewer for pointing out this lack of clarity. We give in the next comment, more details about the different interpretability mechanisms.
>
> > Beyond figures in the paper, can authors make the interpretability claim more rigorous or discuss more about potential implication of such interpretation on training?
>
> Thank you for the suggestion! As stated in the general comment, we are currently designing quantitative evaluations of the proposed perturbations by evaluating the semantic coherence of the proposed approach.
>
> > How does performance vary with teacher quality? If the teacher is changed, does the learned $L$ differ qualitatively in interpretations (e.g., spectrum)?
>
> We thank the reviewer for the relevant idea! One could train Koopman pipelines on teachers at various training stages, analyzing spectral metrics (eigenvalue distributions, mode structures) to reveal phase transitions of training dynamics. Because of the limited timeline and computing capability for revision, we plan to show the results for half-trained teachers on FFHQ and CIFAR-10.
>
> > Do eigen-modes learned on one dataset transfer to others? Any mode-sharing across classes in CIFAR-10?
>
> This is an interesting question! Figure 6 shows the average projection coefficients along modes for many images in the FFQH dataset. This suggests that dataset images share common modes across classes, and that variation comes from class-specific or feature-specific modes. We haven't looked into the overlap between modes across datasets, as MNIST, FFHQ, and CIFAR-10 correspond to substantially different image statistics. However, we acknowledge that our method could allow investigating mode sharing on more similar datasets, and could potentially provide a tool to analyze class-specific image-generative models.
>
> As said above, if you think that we missed an important point in our response and improvement plan, we will be happy to address it during the discussion.
>
> [1] "Bader, P.; Blanes, S.; Casas, F. "Computing the Matrix Exponential with an Optimized Taylor Polynomial Approximation." Mathematics 2019

---

> ### Author Response · Authors · 2025-11-21
>
> ### Concrete Interpretability Mechanisms
>
> - Figure 3 (Dirac perturbations):
>   - With consistency: Perturbations along canonical directions in Koopman space yield localized, plausible, local image transformations. We plan to evaluate the semantic coherence of these transformations. This suggests that we learned a set of Koopman modes that allow us to approximate locally supported functions (like eye attributes) via a finite combination of Koopman modes. As mentioned in the general comment, we will evaluate the semantic coherence of perturbations.
>   - Without consistency: The same perturbations don't induce any local transformations, suggesting our Koopman modes are, interpretability-wise, useless. We can't approximate a locally supported function (like eye attributes) via a finite combination of learned Koopman modes.
> - Spectral signatures (Figs. 4, 6):
>   -  With consistency: We observe smooth eigenvalue decay - which resembles a classic Fourier spectrum. Modes appear to be complex images, and only the sum of all modes contributes to a meaningful image generation. Moreover, steering the generation into the image space unveils that some modes correspond - qualitatively for the moment - to semantic changes, providing insights on the teacher's dynamics.
>   -  Without consistency: We observe a quasi-flat spectrum, except for the first eigenvalues. As suggested by Fig 6, the no-consistency model simply memorizes noise-data mappings, and this can be seen in the modes: they all resemble a face of the dataset. As such, no insights on the learned dynamics are learned during distillation.
> - Downstream robustness (Fig. 7):
>    - Structured basis enables inpainting, super-resolution, and image denoising. This consolidates the idea that our structured Koopman space is well-aligned with the underlying CFM dynamics, as pure distillation fails on such tasks.

---

> > ### Comment · Reviewer_5DZh · 2025-11-27
> >
> > Thank you for your response. I will keep my ratings unchanged unless further experiment results prove otherwise.

---

> ### Author Response · Authors · 2025-12-03
>
> > The paper itself notes challenges scaling to high-resolution images. Results stop at 32×32.
>
> We added 64×64 FFHQ experiments in Section 5.3, where we reached FID of 13.4. The method remains numerically stable, sampling remains one-step, and, more importantly, the Koopman dimension stays the same!.
>
> > Exponentiating a large dense matrix can be costly and numerically delicate.
>
> To follow up on our previous comment, scaling to 64x64 empirically validates the stability of the exponentiation! We also added loss curves (Appendix C) to shed light on the training stability. Despite operating with a 1024x1024 matrix exponentiation, the optimization process behaves smoothly.
>
> > Baselines focus on OT-CFM and Consistency Flow Matching. Missing broader experiment settings on other variants of CFM models or comparisons to SOTA few/one-step approaches.
>
> As requested, we now compare against strong one-step samplers, including Rectified Flow and MeanFlow. This addresses concerns that our method’s performance was not positioned clearly relative to SOTA. Again, we highlight that the main appeal of our method is not the generation quality, but the consistency with the teacher and the interpretability, possible with the Koopman operator decomposition.
>
> > Paper claims a key contribution of interpretability by exposing the Koopman dynamics. Two methods of interpretation are discussed: Dirac delta perturbations and spectral decomposition. However, it's not intuitively clear how such interpretations could shed more insight into flow matching training. Also not clear about the implications of image effects after perturbation in Figure 3.
>
> Your constructive critique actually led us to rewrite Mode decomposition (now in Section 3) and to propose three new interpretability tools in Section 4.4! Notably:
> 1. Indeed, the difference between Dirac and Eigendecomposition is unclear. We instead switched to Schur decomposition to provide a clearer and disentangled decomposition of the Koopman operator. We left the other decomposition examples in the appendix.
> 2. Class-conditioned spectral signature (Fig. 5) that allows us to understand how the generation process changes between two classes. We also show an example of mode addition in Fig 4., that shows how low frequency and high frequency modes compose an image!
> 3. Teacher mode acquisition during training (Fig. 4): Early teachers exhibit almost no usable structure; as training progresses, modes with more negative real parts are learned first.
> 4. Semantic coherence of Koopman modes: We show that we can find some semantically coherent Koopman modes using the consistent model, but not with the pure distilled one.
>
> > Beyond figures in the paper, can authors make the interpretability claim more rigorous or discuss more about potential implication of such interpretation on training?
>
> This has been answered in the last two comments!
>
> > How does performance vary with teacher quality? If the teacher is changed, does the learned $L$ differ qualitatively in interpretations (e.g., spectrum)?
>
> Same!
>
> We would also like to emphasize our last general comment that highlights the new applications for interpretability we propose.

---

### Official Review · Reviewer_12ML · 2025-11-01

**Soundness:** 3
**Presentation:** 3
**Contribution:** 3
**Rating:** 6
**Confidence:** 2

**Summary:**

The paper proposes a method motivated by Koopman theory to design a new training framework that learns an invertible map to lift data into Koopman space, and then learns a linear operator to perform flow in this space. This approach enables faster generation while providing interpretability of the learned transformations.

**Strengths:**

- The paper addresses a common challenge in flow-based models: achieving faster generation while maintaining interpretability.
- The proposed method is well-motivated by solid mathematical theory, grounding the approach in a principled framework. Additionally, leveraging Koopman theory offers a promising avenue for combining efficiency with interpretability, which is a key strength of this work.

**Weaknesses:**

- While the proposed methods are well-motivated, the numerical results seem somewhat limited. The experiments only compare against OT-CFM and Consistency FM, without including many state-of-the-art one-step models such as CTM or MeanFlow. Including these comparisons would better demonstrate the practical advantages of the proposed approach.

- The learned objectives in the proposed method appear quite complex and may be difficult to train in practice. Could the authors provide additional insights or guidance on training stability and implementation challenges?

- Overall, I think this is an interesting paper and provides a promising framework to accelerate generative modeling. However, as mentioned in the conclusion, the current method does not perform well in a complex dataset. This definitely limits its applicability right now.

**Questions:**

I listed some questions directly in weakness.

---

> ### Author Response · Authors · 2025-11-21
>
> > While the proposed methods are well-motivated, the numerical results seem somewhat limited. The experiments only compare against OT-CFM and Consistency FM, without including many state-of-the-art one-step models such as CTM or MeanFlow. Including these comparisons would better demonstrate the practical advantages of the proposed approach.
>
> We thank the reviewer for suggesting relevant baselines. We will add compute-matched timings of all the suggested baselines.
>
> > The learned objectives in the proposed method appear quite complex and may be difficult to train in practice. Could the authors provide additional insights or guidance on training stability and implementation challenges?
>
> We acknowledge that multiple loss terms may raise concerns, but we observed stable training across all three datasets. We provide training details in Appendix 3, including training loss weights. Notably, we weight the losses so that they have similar magnitude:
>
> 1. Phase loss (Eq. 6): 1.0
> 2. Target loss (Eq. 7): 0.01
> 3. Reconstruction loss (Eq. 8): 1.0
> 4. Consistency loss: 1.0
>
> Losses (1, 2, 3) have quite standard implementations, as it is mostly minimize image losses. For the consistency loss, the only implementation trick is to use Pytorch's autograd capabilities to compute the gradient $\nabla_{\boldsymbol{x}} g_{\phi}(x_t) v_t(x_t)$. We recall that thanks to our theorem (Prop. 3), we don't need to simulate full trajectories and can take any $x_t \sim p_t(x_t \vert x_1)$, which is the readily available analytical conditional probability in Conditional Flow Matching pipelines. We plan to add loss training curves to the revision to support this claim.
>
> > Overall, I think this is an interesting paper and provides a promising framework to accelerate generative modeling. However, as mentioned in the conclusion, the current method does not perform well in a complex dataset. This definitely limits its applicability right now.
>
> Thank you for acknowledging our contribution!
> To balance your concern, we would like to emphasize that the complexity scales from MNIST $\to$ FFHQ $\to$ CIFAR. We plan to add experiments on $64\times64$ FFHQ to show that our method also scales in resolution.
>
> Also, as mentioned in the general response above, thanks to the decomposition of the learned Koopman operator, our method provides unprecedented insights and interpretability of the CFM dynamics, and is more than a one-step generation contribution. In this context, relying only on the generation quality (FID) for evaluation doesn't fully encompass the full potential of our method.
>
> As said above, if you think that we missed an important point in our response and improvement plan, we will be happy to address it during the discussion.

---

> > ### Comment · Reviewer_12ML · 2025-11-25
> >
> > Thank you for addressing some of my questions. I will maintain my current scores for now and will continue to check any updates.

---

> ### Author Response · Authors · 2025-12-03
>
> > While the proposed methods are well-motivated, the numerical results seem somewhat limited. The experiments only compare against OT-CFM and Consistency FM, without including many state-of-the-art one-step models such as CTM or MeanFlow. Including these comparisons would better demonstrate the practical advantages of the proposed approach.
>
> The revised paper now includes two baseline families you requested, Rectified Flow and MeanFlow. Table 1 shows compute-matched results, making the performance landscape clearer. However, in addition to achieving fast sampling, our method provides trajectory fidelity, spectral structure, and interpretable dynamics. With this revision, we hope to show our method as a tool for analysis on the Teacher or controlled teacher generation.
>
> > The learned objectives in the proposed method appear quite complex and may be difficult to train in practice. Could the authors provide additional insights or guidance on training stability and implementation challenges?
>
> We added full loss-curve plots and breakdowns in Appendix C, showing stable training dynamics. We'd like to reinstate that thanks to our theoretical results (Proposition 3), all these losses are well aligned given our objective.
>
> > Overall, I think this is an interesting paper and provides a promising framework to accelerate generative modeling. However, as mentioned in the conclusion, the current method does not perform well in a complex dataset. This definitely limits its applicability right now.
>
> We addressed the concerns on the applicability of the method. The revision introduces three major applications directly made possible by time-continuous Koopman linearization. Please see the last global comment. We also want to highlight the ablation, which also shows that latent semantic directions are noticeably better with the consistent model (Fig. 3).

---

### Official Review · Reviewer_EtF6 · 2025-11-01

**Soundness:** 3
**Presentation:** 2
**Contribution:** 2
**Rating:** 4
**Confidence:** 3

**Summary:**

The paper proposes learning a finite-dimensional Koopman linearization for a pre-trained CFM teacher: an encoder/decoder map data Koopman space, and a learned linear generator yields one-step sampling. The key idea is to develope a smulation-free “infinitesimal consistency” loss that aligns the linear dynamics with the teacher’s vector field along the entire path, not just endpoints. The model yeilds competitive FID with 1-step sampling and spectral interpretability of modes.

**Strengths:**

1. Clear problem framing & novelty in training objective. The marginal consistency estimator (Prop. 3) keeps training simulation-free while targeting the true marginal loss, not the conditional surrogate.

2. The interpretability is good. Eigen-analysis and canonical directions provide editable semantics, for example sunglasses in FFHQ.

3. Solid ablation study shows that trajectory fidelity markedly better with consistency loss vs pure boundary distillation.

**Weaknesses:**

1. Many comparisons focus on a distilled ablation and OT-CFM; broader few-step one-step baselines (e.g., strong consistency/rectified variants) and compute-matched timings would strengthen claims.

2. Image domains are small/medium (MNIST, CIFAR-10, 32×32 FFHQ). Add at least a latent-ImageNet or class-conditional setup to test generality.

**Questions:**

1. Does the training on use any stop gradient? How about the computational overhead compared to the simulation-free flow matching, meanflow, consistency flow matching?

---

> ### Author Response · Authors · 2025-11-21
>
> > Many comparisons focus on a distilled ablation and OT-CFM; broader few-step one-step baselines (e.g., strong consistency/rectified variants) and compute-matched timings would strengthen claims.
>
> We thank the reviewer for the relevant baselines. We will add compute-matched timings of all the suggested baselines. However, we would like to emphasize that the two baselines are highly relevant to showcase the superiority of our method, as:
> 1. OT-CFM is the model’s teacher
> 2. Consistency FM has the same ”trajectory fidelity” objective, as opposed to other methods that mainly aim for faster sampling with similar FID.
>
> > Image domains are small/medium (MNIST, CIFAR-10, 32×32 FFHQ). Add at least a
> latent-ImageNet or class-conditional setup to test generality.
>
> We thank the reviewer for this suggestion. We deliberately chose these datasets to disentangle two factors:
> 1. Input dimension (resolution)
> 2. Dataset complexity, where (MNIST $<<$ FFHQ $<<$ CIFAR-10).
> Table 1 results show our method scales across complexity. We also plan to add evaluation on a scaled resolution (64x64) FFHQ dataset. However, we won't be able to train a model on ImageNet because of limited compute capabilities, and given the numerous experiments we are running to improve the paper!
>
> Also mentioned in the general response above, thanks to the decomposition of the learned Koopman operator, our method provides unprecedented insights and interpretability of the CFM dynamics, and is more than a one-step generation contribution. In this context, relying only on the generation quality (FID) for evaluation doesn't fully encompass the full potential of our method.
>
> > Does the training use any stop gradient?
>
> Good question! No, the training doesn't use any stop gradient. To compute the $\mathcal{L}_ {\text{cons}}$ loss, we use Pytorch's autograd capabilities. The network's gradients relative to the final loss, are computed using the usual Pytorch's backpropagation (so for $\mathcal{L}_{\text{cons}}$, we backpropagate twice).
>
> > How about the computational overhead compared to the simulation-free flow matching, meanflow, consistency flow matching?
>
> We thank the reviewer for suggesting relevant baselines. We will add compute-matched timings of all the suggested baselines.
>
> As said above, if you think that we missed an important point in our response and improvement plan, we will be happy to address it during the discussion.

---

> ### Author Response · Authors · 2025-12-03
>
> > Many comparisons focus on a distilled ablation and OT-CFM; broader few-step one-step baselines (e.g., strong consistency/rectified variants) and compute-matched timings would strengthen claims.
>
> As suggested, the revised paper expands Table 1 with Rectified Flow and MeanFlow variants, all trained under the same preprocessing and measured in matched wall-clock time. These results now clarify where Koopman-CFM stands relative to strong one-step and few-step models. Importantly, the new revision highlights the conceptual distinction of our method from baselines. In addition to achieving fast sampling, ours provides trajectory fidelity, spectral structure, and interpretable dynamics. With this revision, we hoped to show our method as a tool for analysis on the Teacher or controlled teacher generation.
>
> > Image domains are small/medium (MNIST, CIFAR-10, 32×32 FFHQ). Add at least a latent-ImageNet or class-conditional setup to test generality.
>
> We now include FFHQ-64x64 experiments (Section 5.3), demonstrating that Koopman linearization scales with both dimension and dataset complexity.
>
> > Does the training use any stop gradient?
>
> We added full loss-curve plots and breakdowns in Appendix C, showing stable training dynamics.
>
> We would also like to emphasize our last general comment that highlights the new applications for interpretability we propose.

---

### Author Response · Authors · 2025-11-21

We sincerely thank all reviewers for their time and insightful, professional feedback on our submission. We also thank our AC and SAC for coordinating the discussion process.

We are highly encouraged that all reviewers recognize the **novelty and solid mathematical foundation** of our work, with ratings at or near the acceptance threshold (4-6):

## Recognized Strengths
1. **Novel contribution**: "Clear problem framing \& novelty" (EtF6), "solid mathematical theory" (12ML), "interesting...clean construction" (5DZh), "novel algebraic perspective" (mSkg)
2. **Interpretability**: "Good interpretability" (EtF6), eigenmodes "uncover semantically meaningful latent directions" (mSkg), "promising avenue for combining efficiency with interpretability" (12ML)
3. **Theoretical soundness**: Good (EtF6, 12ML, 5DZh, mSkg) soundness and presentation
4.  **Practical benefits**: "Wall-clock improvements" (5DZh), "significant speedups" (EtF6)

The main concerns raised are:
- **Limited baselines** and experimental scope
- **Scalability** to high-resolution images
- **Interpretability validation** with quantitative experiments and some points lacking clarity.

We address these systematically below and plan to update our submission accordingly.

Before, we would like to first emphasise our contributions again.

### How is our approach fundamentally different from distillation techniques?

We emphasise a fundamental distinction between our method and prior works.

Current related work on few-step distillation focuses only on reducing the time-steps, by distilling the trajectories into straighter paths (e.g. Rectified Flow) or by directly learning one-step generation (e.g. CTM, Mean Flow).

As they only focus on reducing the number of timesteps, they cannot provide interpretability or insights on the teacher's dynamics, and work as black-box sampling speed improvements.

**Our method distinguishes** by (1) *enforcing trajectory consistency* with the **teacher's time-continuous dynamics** and (2) *globally linearizing* the **full evolution**. This ensures the learned Koopman operator aligns with the underlying generative process, **enabling analysis and interpretability**, unavailable in prior work.

**Three capabilities unavailable in prior work**
1. **Diagnostic insights**: Removing $\mathcal{L}_{\text{cons}}$ improves FID (7.5 vs 10.1) but degrades trajectory fidelity 260$\times$ (MSE: $5 \times 10^{-6}$ vs $1.3 \times 10^{-3}$). This confirms that the *pure one-step distillation* ignores generation dynamics. In this case, the distillation training places all the strain on the encoder-decoder that simply memorizes noise-data pairings.
2. **Interpretability**: *Eigendecomposition* $L = P\Lambda P^{-1}$ exposes growth rates, oscillations, and semantic modes (Figs 6, 9) of the CFM dynamics. Similarly, *Dirac perturbations* allow us to probe if the modes we learn via the Koopman framework allow, through combination, local modification in images. Perturbation along modes suggests, qualitatively for now, that modes may align with semantic features.
3. **Analytical sampling**: One-step generation via $\exp(L)$ achieves 20-100$\times$ speedup (Table 1). While one-step generation is not a novel objective, it is the first time that it can be achieved with a **global operator** encapsulating **full trajectory dynamics**.


**FID is insufficient** to evaluate the quality of the learned model. Indeed, no baseline offers trajectory reproduction and analysis of **continuous-time dynamics**, along with improved sampling speed.


### Why This Problem is Non-Trivial

Applying continuous Koopman theory to CFM dynamics requires solving three fundamental challenges:

- **Non-autonomous dynamics**:  CFM velocity $v_t(x_t)$ is *time-dependent*, thus non-autonomous, Koopman theory can't be applied straightforwardly. We use an affine lift (Sec 4.1, Theorem 1) to transform dynamics into Koopman-compatible, autonomous dynamics.
 - **Tractable training**: A naïve use of Koopman theory would require to *simulate full trajectories*. Thanks to our **Marginal consistency estimator**, formulated as our loss $\mathcal{L}_\text{cons}$ (Sec 4.3, Props 2-3), we provide a tractable training avoiding the need to simulate full trajectories.
- **Identifiable coordinates**: Koopman theory is gauge-invariant (we can find multiple coordinate systems and Koopman operators satisfying the solution). By breaking the gauge freedom (Sec 4.2, Corollary 1.1), $\mathcal{L}_\text{rec}$ allows us to find a tractable Koopman operator.

**Main results**: Our work provides the first **global linearization** of continuous-time generative flows with **simulation-free training**.

---

### Author Response · Authors · 2025-11-21

## Commitments for Revision

We plan to add, by the end of the coming week, the following experiments to strengthen the paper, in accordance with reviewers' suggestions.

1. **Extended baselines**: We plan to add Rectified Flow, CTM, MeanFlow, to the comparisons in Table 1.
2. **Scalability**: Our paper shows scalability from relatively simple (MNIST) to more complex (FFHQ and then CIFAR-10) image datasets, with fixed resolution. To show scalability in resolution, we will add FFHQ results on 64x64.
3. **Interpretability**: We plan to evaluate quantitatively the semantic coherence of the discovered modes.
4. **Clarity** We will improve the presentation of the method, particularly for interpretability (section 4.4).

## Summary of our answer

- **Core contribution**: We provide the first tractable framework for globally linearizing CFM dynamics using Koopman operator theory, enabling interpretion of CFM dynamics using mode decomposition of the operator, and one-step analytical sampling.

- **Key innovations**: (1) Time-augmented Koopman framework for non-autonomous dynamics (Theorem 1), (2) Simulation-free marginal consistency loss enforcing full-trajectory fidelity (Props 2-3), (3) Mode decomposition revealing semantic structure.

- **Reviewer consensus**: The reviewers recognize the novelty of the approach, principled in solid theory with a rigorously proved method. Concerns center on baselines, scalability, and interpretability validation.

- **Our response**: We commit to broader baselines (Rectified Flow, CTM, MeanFlow), 64$\times$64 results, and semantic validation of the interpretability to answer reviewers' concerns.

We address each reviewer's specific concerns in detail below, providing additional discussions and committing to new experiments where appropriate.

We hope those first comments will start fruitful discussions to improve the paper. If you think that we missed an important point in our response and improvement plan, we will be happy to address it during the discussion.

---

### Author Response · Authors · 2025-12-03
**Revision and comment for new AC.**

Dear Area Chair,

Thank you for overseeing the discussion of our submission. Below, we concisely summarize the reviewers’ assessments and the concrete revisions we implemented to address their concerns.

Across all reviewers, there was **clear and consistent recognition of the core strengths of our work**. Reviewers highlighted that the paper introduces a **principled and novel formulation** grounded in solid mathematical theory, especially the marginal consistency estimator (Prop. 3), which preserves the simulation-free nature of CFM while targeting the *true* marginal objective.
Reviewers (mSkg), (5DZh), and (12ML) emphasized the **interpretability** unlocked by global linearization—describing it as an “interesting” direction, a “promising avenue for combining efficiency with interpretability,” and a meaningful contribution beyond standard distillation pipelines. Reviewer (5DZh) also requested more rigor behind the interpretability claims; our revision addresses this with new quantitative experiments that demonstrate the semantic coherence of Koopman modes.

Below we summarize major concerns and how the revision directly resolves them.

### Limited baselines: (EtF6, 12ML, 5DZh)

Reviewers requested broader comparisons against strong one-step or few-step baselines, along with clearer timing analysis.

**Revision:**
We expanded Table 1 to include Rectified Flow and MeanFlow under compute-matched settings.
Our method remains competitive in FID while achieving **20–100× faster sampling**,  and remains the *only* approach offering both **trajectory fidelity** and **interpretable Koopman spectra**, a distinction repeatedly acknowledged by reviewers.

### Scalability: (EtF6, 12ML, 5DZh)

Reviewers asked whether our method scales to higher-resolution data and more complex distributions.

**Revision:**
We added experiments on **FFHQ-64×64** reaching an FID of 13.4 for our model, demonstrating scalability on **resolution** - completing our findings on **dataset complexity**. We emphasize that this experiment shows that matrix exponentiation is *not* a conceptual bottleneck, as the Koopman dimension does not scale monotonically with input resolution. Thus, we believe this concern is mainly engineering-oriented rather than a fundamental limitation.

### Interpretability validation (5DZh, mSkg)
Reviewers found the interpretability aspects compelling but requested stronger justification and clearer examples.

**Revisions:**
We **clarified** the presentation of interpretability:
1. We first propose a unified decomposition (Schur decomposition, section 3) to uncover Koopman modes, instead of Dirac and Eigendecomposition. This allows for easier reading, as (5DZh) highlighted that presenting two methods was akin to misunderstanding
2. We rewrote Section 4.4 and proposed (1) quantitative methods to interpret CFM dynamics (2) clearer interpretability applications from our method.

Namely, our key applications are:
- **Class-conditioned spectral signature** As highlighted by (EtF6), no class condition analysis had been provided. We define class transfer functions that highlight differences in the image generation process from one class to another.
- **Finding semantically coherent modes** We define a semantic coherence measure that allows us to find semantically coherent modes. We show that only the consistent model is able to provide semantically coherent modes for different image attributes, validating our global approach.
- **Teacher-training analysis.** Using our framework, we demonstrate how a CFM teacher acquires Koopman modes over training. As Figs.~4a--4b show, early checkpoints contain almost no coherent structure, while mid-training begins to acquire low-real-part modes—addressing the reviewer’s question regarding the implications of the Koopman decomposition.

### Minor concerns: Training stability and implementation clarity: (12ML, mSkg)
The reviewers requested clarity on loss weighting, stability, and autodiff details.

**Revision:**
We added explicit descriptions of loss weighting, implementation considerations, and explained how autodiff is applied in the consistency objective. Training curves across datasets show **stable and smooth optimization**.

Most importantly, our revision highlights the broader value of **global linearization** going beyond raw FID metrics: our approach not only enables fast sampling but also unlocks **new analytical tools**, **interpretable spectral decompositions**, and **fine-grained control** over the teacher CFM model. These capabilities significantly expand the applicability and provide an entirely novel mathematically-principled toolbox for understanding the dynamics of CFM-based generative models.

Lastly, we emphasize that will provide a complete implementation of our approach to enable full reproducibility of all of our results and to facilitate follow-up work by the community.

---

### Author Response · Authors · 2025-12-03
**Additional note: : New Applications Enabled by Global Koopman Linearization**

The revision adds three new applications demonstrating the practical benefits of time-continuous Koopman linearization:

1. **Semantically meaningful editing directions** (Fig. 4): Single eigenmodes reliably capture semantic concepts (e.g., sunglasses), achieving up to **0.97 semantic coherence** (defined in section 4.4) —a capability not available in prior distillation methods.
2. **Class conditional spectral signatures** (Fig 6): We provide a spectral analysis of class conditioned datasets. We show that on CIFAR-10, common modes corresponds to lower frequencies (small $Re(\lambda)$) and discrepancies happen in high frequencies (high $Re(\lambda)$)
3. **Teacher mode acquisition over training** (Results in appendix E.2):  We show how CFM models progressively acquire Koopman structure, by comparing Koopman modes of lightly trained teachers to the modes of fully trained teachers. We discover that coherent modes emerging only mid-training.

Thank you again for your consideration.

Thanks and best regards,

Authors of Submission 14319

---

### Note · Program_Chairs · 2026-01-17
**Submission Desk Rejected by Program Chairs**

The following references in this submission do not refer to real documents and/or have major errors in bibliographic information:

 ang Song, Prafulla Dhariwal, Mark Zhang, and Karsten Kreis. Consistency models. In International Conference on Machine Learning (ICML), 2023.
Yujia Liu, Chuan Guo Li, Kuan-Chieh Zhou, and Anima Anandkumar. Flow matching with stochastic differential equations. arXiv preprint arXiv:2306.02393, 2023b.
Simian Luo, Yiqin Wu, Surui Wang, Puchao Chen, Shijie Zhao, Jun Zhu, et al. Latent consistency models: Synthesizing high-resolution images with few-step inference. arXiv preprint arXiv:2310.04378, 2023.
Aram-Alexandre Pooladian, Alexander Gushchin, Regina Barzilay, and Tommi Jaakkola. Multisample flow matching: Straightening flows with minibatch couplings. arXiv preprint arXiv:2305.17160, 2023.
Alexander Tong, Nikolay Malkin, Guillaume Huguet, Yanlei Zhang, Jarrid Liu, Kilian Rector-Brooks, Guy Wolf Fatras, Elizabeth Creager, and Yoshua Bengio. Conditional flow matching: Simulation-free dynamic optimal transport. arXiv preprint arXiv:2302.00482, 2023.
Yaron Lipman, Ioannis Gkioulekas, Tatsunori Hashimoto, William T Liu, Ben Poole, Ricky Richter-Powell, Robin Rombach, Ali Toker, and Jiaxin Wu. Flow matching for generative modeling. In International Conference on Learning Representations (ICLR), 2023.